# DurMI: Duration Loss as a Membership Signal in TTS Models

## Abstract

Text-to-speech (TTS) models such as FastSpeech2, Grad-TTS, and VITS2 achieve state-of-the-art quality but risk memorizing and leaking sensitive training data. Existing membership inference attacks (MIAs) for diffusion-based TTS systems typically rely on denoising errors, which are costly to compute and often weak at capturing sample-specific memorization.

We introduce DurMI, the first membership inference attack that exploits duration loss—a core alignment signal in modern TTS pipelines—as a highly discriminative indicator of membership. Duration loss reflects the model's tendency to overfit alignment targets, whether obtained from deterministic aligners (e.g., MAS, MFA) or stochastic predictors (e.g., VITS2), enabling accurate inference with a single forward pass. Beyond this family of systems, we further show that DurMI can be extended to alignment-free and zero-shot TTS models via proxy indicators derived from length discrepancies, broadening the attack surface to emerging architectures.

Experiments on TTS systems, such as Grad-TTS, WaveGrad2, VoiceFlow, FastSpeech2, and VITS2, demonstrate that DurMI consistently performs better than earlier MIAs, especially on waveform-level synthesis where current attacks are inadequate. We further assess DP-SGD as a defense and discover that DurMI endures even in the presence of substantial noise, underscoring the need for stronger, TTS-specific privacy safeguards. These findings show DurMI's efficacy, efficiency, and wide range of applications.

## 1 Introduction

Research in text-to-speech (TTS) has moved quickly, and recent systems can produce speech that is often difficult to distinguish from real recordings. Grad-TTS (Popov et al., 2021), WaveGrad2 (Chen et al., 2021), VoiceFlow (Guo et al., 2024), FastSpeech2 (Ren et al., 2020), and VITS2 (Kong et al., 2023b) are known techniques that contribute to this progress. These models are trained on large-scale datasets that often contain sensitive or proprietary content, raising critical concerns about privacy leakage. In applications like voice assistants or medical TTS, where the training data may reveal personal identification, health-related information, or even geographical cues (Chen et al., 2023), these concerns become very severe.

MIAs have been extensively studied in computer vision (Chen et al., 2020; Carlini et al., 2022; Li et al., 2024b) and natural language processing (Shi et al., 2023; Mattern et al., 2023; Fu et al., 2024). They have also been adapted to generative models, such as GANs and VAEs (Hayes et al., 2017; Hilprecht et al., 2019; Sui et al., 2023), as well as more recent diffusion-based models (Matsumoto et al., 2023; Duan et al., 2023; Hu & Pang, 2023).

To improve efficiency, Proximal Initialization Attack (PIA) (Kong et al., 2023a) reduces the number of denoising steps and extends MIA to mel-spectrogram and waveform-level TTS models. However, these approaches treat TTS largely as a generic generative model, overlooking architectural features that are central to speech synthesis. In particular, *alignment mechanisms and duration supervision* are unique to TTS pipelines and directly influence how models memorize training utterances, yet remain unexploited by prior MIAs.

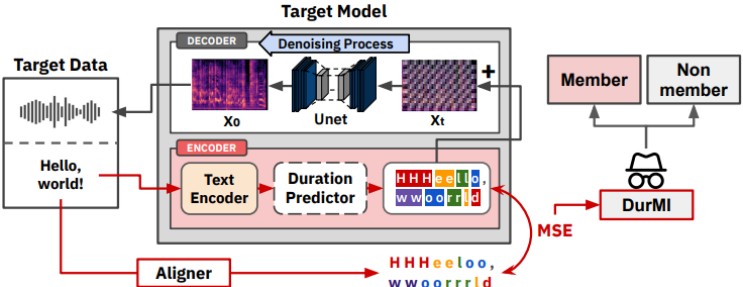

Figure 1: Overview of DurMI: illustrated here on a diffusion-based TTS model, where the difference between predicted and ground-truth durations from the aligner is used as a membership signal. DurMI, however, applies broadly across transformer-, flow-matching-, and stochastic-alignment models as well. It requires only a single forward pass up to the decoder stage (red arrow).

**In this work, we propose *DurMI* (Duration Loss-Based Membership Inference), the first attack to exploit *duration loss* as a discriminative signal for membership inference.** Our key insight is that duration predictors are trained to match sample-specific alignment targets, whether deterministic (e.g., MAS (Kim et al., 2020), MFA (McAuliffe et al., 2017)) or stochastic as in VITS2 (Kong et al., 2023b), which encourages overfitting to utterance-level timing patterns. This makes duration loss a highly effective signal for membership inference. As shown in Figure 1, DurMI requires only a single forward pass before the decoder, bypassing diffusion entirely and yielding **50–100× speedups** over prior diffusion-based MIAs.

Our study operates in a grey-box setting, in that we access internal signals (e.g., duration loss) but not model weights or architectural details, following the standard evaluation protocol in MIA (Duan et al., 2023; Kong et al., 2023a). This assumption reveals worst-case vulnerabilities critical for privacy auditing and defense design, and reflects realistic contexts such as internal audits, regulatory compliance, or fine-tuning on user data.

We evaluate DurMI on Grad-TTS, WaveGrad2, VoiceFlow, FastSpeech2, and VITS2 across three benchmark datasets (LJSpeech (Ito & Johnson, 2017), VCTK (Yamagishi et al., 2019), and LibriTTS (Zen et al., 2019)). Through these experiments, we establish DurMI's key contributions: (i) it consistently outperforms prior MIAs in terms of detection accuracy, (ii) it generalizes across diverse architectures, including VITS2 with stochastic alignment, (iii) it achieves high efficiency, requiring only a single forward pass (over 100× faster than SecMI), and (iv) it is modality-agnostic, applying equally well to spectrogram- and waveform-based synthesis.

Finally, while zero-shot and alignment-free TTS has emerged, recent studies show these models often degrade on complex text and reintroduce auxiliary alignment for intelligibility (Jiang et al., 2025; Neekhara et al., 2024), and we provide a detailed discussion in Appendix A.14. Together with the continued deployment of duration-supervised systems in industry, this indicates that alignment remains central to current and near-future TTS. Although DurMI cannot be directly applied to alignment-free architectures, our experiments with proxy indicators show that they remain moderately effective, demonstrating a feasible path for extending membership inference beyond explicitly aligned systems.

Furthermore, our DP-SGD defense studies demonstrate that DurMI performance endures even in the face of significant noise, underscoring the necessity for more sophisticated and alignment-agnostic protection strategies.

## 2 RELATED WORK

**Membership Inference Attacks.** Membership inference attacks (MIAs) aim to determine whether a data sample was part of a model's training set. First studied in discriminative models via output confidence scores (Shokri et al., 2017), they have since been extended to generative settings. LOGAN (Hayes et al., 2017) showed leakage in GANs, and recent work in LLMs uses

Table 1: Comparison of existing membership inference attacks targeting diffusion models.

| Method | Key Idea | Sampling | Computational Cost |
|--------|----------|----------|--------------------|
| Naive Attack | Computes MSE between predicted and ground-truth noise | Stochastic (DDPM) | Moderate (1 query) |
| SecMI | Collects timestep-wise noise prediction errors for classification | Deterministic (DDIM) | High (full trajectory) |
| PIA | Reconstructs sample via DDIM, re-diffuses once to measure error | Deterministic (DDIM) | Moderate (1–2 queries) |
| PIAN | Normalized variant of PIA using $L_1$ norm | Deterministic (DDIM) | Moderate (1–2 queries) |

log-likelihoods of rare tokens as membership signals (Shi et al., 2023; Zeng et al., 2023; Carlini et al., 2021).

For diffusion models, MIAs exploit reconstruction errors during denoising. The Naive Attack (Matsumoto et al., 2023) measures mean squared error between predicted and ground-truth noise. SecMI (Duan et al., 2023) aggregates per-timestep errors, improving accuracy under deterministic DDIM sampling but at high cost. PIA and PIAN (Kong et al., 2023a) reduce overhead by reconstructing samples with few steps, though with weaker robustness. Table 1 summarizes these methods.

These techniques are largely tailored to images and overlook signals unique to TTS. Grey-box analysis, although not always feasible in deployment, remains standard in MIA (Duan et al., 2023; Kong et al., 2023a) and is critical for revealing worst-case vulnerabilities that guide auditing and defenses.

**Text-to-Speech Models.** Early autoregressive models such as WaveNet (van den Oord et al., 2016) and Tacotron (Wang et al., 2017) achieved high fidelity but suffered from slow inference. Non-autoregressive (NAR) models like FastSpeech2 (Ren et al., 2020) introduced explicit duration predictors for parallel and controllable synthesis, and duration-based supervision is now standard across many NAR systems. Our work is the first to show that this alignment loss itself leaks membership information.

Grad-TTS (Popov et al., 2021) extended TTS with diffusion-based mel-spectrogram generation, while WaveGrad2 (Chen et al., 2021) synthesized raw audio directly. Building on diffusion, flow-matching approaches (Mehta et al., 2024; Chen et al., 2024) improve stability and sampling efficiency; for example, VoiceFlow (Guo et al., 2024) enables faster and more robust synthesis. VITS2 (Kong et al., 2023b) further enhances VITS (Kim et al., 2021) through stochastic duration modeling, supporting more diverse prosody and improved naturalness.

Alignment-free and zero-shot systems such as E2-TTS (Eskimez et al., 2024) and F5-TTS (Chen et al., 2024) have recently emerged, though their robustness remains debated. MegaTTS 3 (Jiang et al., 2025) and NVIDIA's T5-TTS (Neekhara et al., 2024) both report quality gains by reintroducing explicit or monotonic alignment, suggesting that alignment will remain important. Although DurMI cannot be directly applied to zero-shot models, these systems still exhibit implicit alignment cues—such as discrepancies between target and generated utterance length—which we highlight as potential *proxy indicators* for extending MIA to alignment-free TTS (see Section 6 and Appendix A.7); moreover, our experiments using these proxies confirm that they provide moderate but meaningful membership inference performance.

## 3 PRELIMINARIES

This section reviews the role of duration loss and alignment strategies in representative TTS architectures: Grad-TTS, WaveGrad2, VITS2, VoiceFlow, and FastSpeech2.

### 3.1 DURATION LOSS

A large class of modern TTS models generally consist of three modules: a text encoder, a duration predictor, and a decoder. While auxiliary objectives such as pitch or energy vary across systems,

*duration loss is universally present.* It supervises phoneme-to-frame alignment, enforcing sample-specific timing patterns. DurMI builds directly on this shared mechanism: by comparing predicted and ground-truth durations, it extracts membership signals that generalize across diverse architectures.

**Grad-TTS.** Durations $d \in \mathbb{R}^L$ are obtained via MAS, which computes sample-specific phoneme-to-frame mappings. The model minimizes

$$\mathcal{L}_{\mathrm{dur}}^{\mathrm{GT}} = \|f_{\mathrm{dur}}(\mathrm{sg}[f_{\mathrm{enc}}(c)]) - d\|_2 \,, \tag{1}$$

where $\mathrm{sg}[\cdot]$ blocks gradients to the encoder. Because MAS adapts dynamically during training, it introduces variability across utterances, slightly weakening membership leakage but improving synthesis quality.

**WaveGrad2.** Here, durations $d$ are precomputed using MFA, a fixed alignment tool. The predictor minimizes

$$\mathcal{L}_{\mathrm{dur}}^{\mathrm{WG}} = \|\log \hat{d} - \log d\|_2. \tag{2}$$

Unlike MAS, MFA provides static, non-adaptive alignments, which tend to overfit to training utterances. This makes duration loss in WaveGrad2 a stronger membership signal.

**VITS2.** VITS2 combines MAS-based targets with adversarial learning. Predicted durations $\hat{d}$ are optimized as

$$\mathcal{L}_{\mathrm{dur}}^{\mathrm{V2}} = \mathrm{MSE}(\hat{d}, d) + \lambda L_{\mathrm{adv}}(G), \tag{3}$$

where $L_{\mathrm{adv}}$ encourages natural duration distributions. This stochastic training setup reduces determinism but still preserves sample-specific information, showing that DurMI generalizes beyond purely deterministic predictors.

**VoiceFlow and FastSpeech2.** Both rely on forced alignments (e.g., MFA) to generate ground-truth durations and minimize a mean squared error:

$$\mathcal{L}_{\mathrm{dur}}^{\mathrm{VF,FS2}} = \frac{1}{N} \sum_{i=1}^{N} \|\hat{d}_i - d_i\|^2. \tag{4}$$

These models provide stable and explicit supervision, making duration loss a reliable membership signal.

### 3.2 Alignment Mechanisms for Duration Prediction

**Montreal Forced Aligner (MFA).** MFA is an offline, non-differentiable aligner based on Gaussian Mixture Model–Hidden Markov Model (GMM-HMM) acoustic models and MFCC features, implemented in Kaldi. It produces fixed phoneme-to-frame alignments independent of model parameters and is widely used in TTS pipelines. These static alignments often amplify memorization signals.

**Monotonic Alignment Search (MAS).** MAS is a differentiable dynamic programming algorithm that finds monotonic text–audio alignments by maximizing cumulative likelihood:

$$Q_{i,j} = \max(Q_{i-1,j-1}, Q_{i,j-1}) + \log \mathcal{N}(z_j; \mu_i, \sigma_i),$$

where $z_j$ is an acoustic frame and $(\mu_i, \sigma_i)$ are phoneme-level Gaussian parameters. Backtracking recovers the alignment path $A^*$, and durations are computed as

$$d_i = \log \left( \sum_{j=1}^{F} \mathbb{I}\{A^*(j) = i\} \right).$$

Unlike MFA, MAS integrates alignment into training, yielding more adaptive but less deterministic targets, which weakens overfitting signals.

**Stochastic Duration Modeling.**    VITS2 introduces stochastic duration prediction to capture natural variability in rhythm and prosody. Durations are generated as

$$\hat{d} = G(z_d, h_{\text{text}}), \quad z_d \sim \mathcal{N}(0, I),$$

where $G$ is trained with a combination of mean squared error and adversarial loss:

$$\mathcal{L}_{\text{dur}}^{\text{V2}} = \text{MSE}(\hat{d}, d) + \lambda L_{\text{adv}}(G).$$

This formulation reduces determinism but still preserves sample-specific timing, showing that alignment supervision remains embedded even in stochastic predictors.

## 4 DURATION LOSS-BASED MEMBERSHIP INFERENCE

We introduce DurMI, a grey-box membership inference attack that leverages *duration loss* as a discriminative signal in TTS models. Our key insight is that duration predictors are trained to minimize sample-specific alignment errors – leading to potential overfitting – which can be exploited for identifying training membership.

### 4.1 THREAT MODEL AND ASSUMPTIONS

We assume a grey-box threat model in which the adversary has full access to a trained TTS model, including the encoder $f_{\text{enc}}$, duration predictor $f_{\text{dur}}(\cdot; \theta)$, and the loss function $\mathcal{L}_{\text{dur}}$. Given a target sample $x = (c, a)$ and its alignment target $d$ (obtained from MAS or MFA), the goal is to determine whether $x$ belongs to the training set $\mathcal{D}_{\text{train}}$.

### 4.2 FORMULATION OF DURMI

Let $f_{\text{enc}}(c)$ denote the phoneme-level representation of the input text sequence $c$, and let $f_{\text{dur}}(\cdot; \theta)$ be the duration predictor parameterized by $\theta$. The duration loss for input $x$ is computed as:

$$\mathcal{L}_{\text{dur}}(x; \theta) = \|f_{\text{dur}}(\text{sg}[f_{\text{enc}}(c)]; \theta) - d\|_p, \tag{5}$$

where $d$ is the ground-truth log-duration vector and $\text{sg}[\cdot]$ is the stop-gradient operator that blocks gradients during optimization. The norm $\|\cdot\|_p$ is selected by the attacker (typically $p = 2$).

The adversary then defines a binary membership function $\mathcal{M}$ based on thresholding the loss:

$$\mathcal{M}(x) = \begin{cases} 1 & \text{if } \mathcal{L}_{\text{dur}}(x; \theta) < T \\ 0 & \text{otherwise} \end{cases} \tag{6}$$

where $T$ is a decision threshold estimated via a calibration set or a shadow model. In practice, the attacker computes the empirical loss distributions for member and non-member samples and selects (T) to maximize separation (e.g., maximizing TPR@1%FPR or selecting the ROC-operating point that optimizes attack utility). Since duration predictors often overfit to deterministic alignment targets, training samples tend to exhibit lower loss values, making threshold-based membership prediction effective.

### 4.3 COMPARISON WITH DIFFUSION-BASED MIAS

Table 2 compares DurMI against existing MIA techniques that rely on diffusion loss or timestep-wise noise prediction errors. While these prior approaches are applicable to generic diffusion models, they often suffer from high computational cost and require fine-grained calibration. In contrast, DurMI offers a TTS-specific yet efficient and highly discriminative alternative.

Table 2: Comparison of duration loss (DurMI) and diffusion-based membership signals.

| Aspect | DurMI | Diffusion-based MIAs |
|---|---|---|
| Target signal | Duration loss | Noise prediction error |
| Sample specificity | High | Low to moderate |
| Loss variance | Low | High |
| Computational cost | Low (single forward pass) | High (multi-step rollouts) |

Table 3: Comparison of intra-class variance, inter-class variance, and LDA scores for duration loss (DurMI) and diffusion loss (Naive Attack, PIA) used as membership signals.

| Method | Intra-class variance | Inter-class variance | LDA |
|---|---|---|---|
| DurMI | **0.002** | 0.012 | **6.0** |
| Naive Attack | 6.658 | 0.031 | 0.004 |
| PIA | 27.549 | 0.203 | 0.007 |

## 4.4 ADVANTAGES OF DURATION LOSS FOR MEMBERSHIP INFERENCE

Duration loss offers some important distinctions from previous MIAs on diffusion models (Matsumoto et al., 2023; Duan et al., 2023; Kong et al., 2023a), which solely concentrate on reconstruction loss or noise prediction errors as membership signals.

First, duration loss is sample-specific, whether derived from deterministic aligners (e.g., MAS, MFA) or stochastic predictors (e.g., VITS2). Deterministic alignments encourage exact overfitting to utterance-level timing, while stochastic predictors still rely on training-conditioned distributions that retain sample-level bias. In both cases, the duration loss tightly encodes alignment signals tied to individual training examples, unlike diffusion loss, which is distributional and exhibits higher intra-class variance.

Second, duration loss exhibits stronger separability.As shown in Table 3, duration loss has significantly lower intra-class variance and higher inter-class separability, as measured by Fisher's Linear Discriminant Analysis (LDA) score, a metric that captures how well two classes (member vs. non-member) are separated based on the ratio of between-class to within-class variance.This improved separation effectively supports simple and effective threshold-based inference.

Figure 2 further presents the distributional gap between member and non-member samples across various MIA techniques. DurMI exhibits sharper decision margins for LJSpeech and LibriTTS, while diffusion-based losses exhibit substantial overlap, particularly for LJSpeech. DurMI shows greater overlap on the VCTK dataset, which is discussed in more detail in Appendix A.6.

Finally, DurMI is computationally efficient. Unlike prior attacks that require full diffusion rollouts or multiple inference passes (e.g., SecMI, PIA), DurMI computes a single forward pass prior to the decoder, independent of decoder-level sampling. This makes DurMI particularly practical for large-scale TTS systems.

## 5 EXPERIMENTS

We evaluate DurMI on five representative TTS architectures: Grad-TTS, WaveGrad2, FastSpeech2, VoiceFlow, and VITS2. Experiments are conducted on three widely used benchmarks: LJSpeech (Ito & Johnson, 2017), VCTK (Yamagishi et al., 2019), and LibriTTS (Zen et al., 2019). Each experiment is repeated three times, and averages are reported, as standard deviations are consistently below 0.1%. Baselines include Naive Attack, SecMI, PIA, and PIAN and full implementation details and preprocessing procedures provided in Appendix A.1.

To ensure fair evaluation, we split each dataset evenly into member samples (50%, used for training) and non-member samples (50%, held out). From both pools, 20% are further used as a calibration set and 80% as an evaluation set. The calibration set assumes that the adversary has access to a small subset of both member and non-member data, as is standard in MIA research, and is used solely to

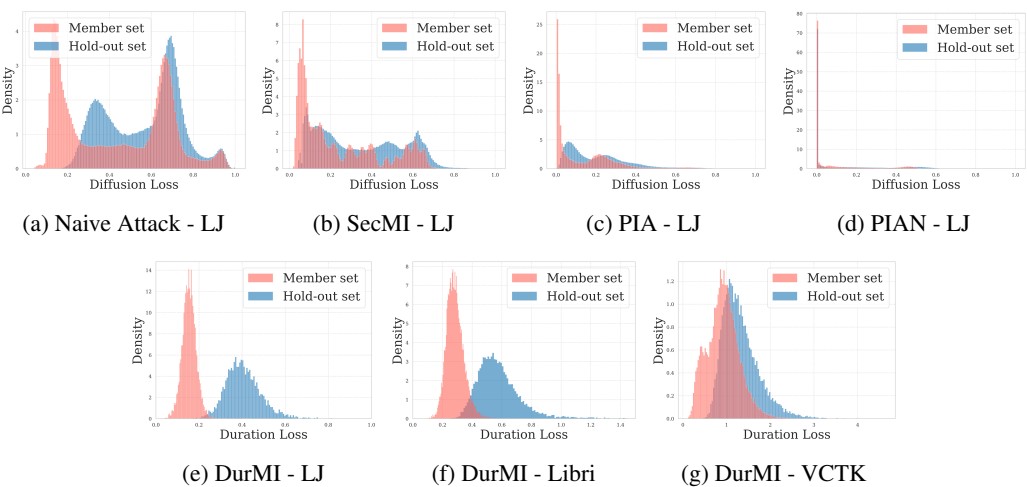

Figure 2: Member vs. Non-member distribution separability using diffusion loss (Naive, SecMI, PIA, PIAN) vs. duration loss (DurMI) across datasets: LJSpeech (LJ), LibriTTS (Libri), and VCTK.

Table 4: Performance of MIA methods on GradTTS across various datasets.

| | LJSpeech | | LibriTTS | | VCTK | |
|---|---|---|---|---|---|---|
| | AUC | TPR@1% FPR | AUC | TPR@1% FPR | AUC | TPR@1% FPR |
| Naive Attack | 86.7 | 55.0 | 94.5 | 58.1 | 73.2 | **29.5** |
| SecMI | 94.4 | 70.3 | 90.2 | 55.2 | 72.8 | 8.1 |
| PIA | 89.0 | 55.0 | 89.3 | 47.0 | 64.4 | 9.7 |
| PIAN | 69.0 | 37.4 | 81.8 | 37.4 | 66.6 | 6.1 |
| DurMI | **99.7** | **99.1** | **98.9** | **82.8** | **86.7** | 18.2 |

set decision thresholds. Evaluation samples are never used for calibration, ensuring that AUROC and TPR@1% FPR reflect unbiased attack performance.

We report two standard metrics used in membership inference literature (Carlini et al., 2022; 2023; Li et al., 2024a). The first is AUC which measures the overall ability of the attack to distinguish members from non-members. The second metric is the series of TPRs at 1, 0.1, and 0.01% FPRs, which quantifies true positive rates under severe false-positive limitations and demonstrates precision in privacy-sensitive circumstances. Appendix A.10 contains the results for TPRs at 0.1 and 0.01% FPRs.

**Compared to Baselines.** DurMI consistently outperforms baseline MIA methods across models and datasets, achieving the highest AUC and TPR@1%FPR as shown in Tables 4 and 6 with the exception of Grad-TTS on VCTK set, which we analyze in detail later. ROC curves in Figure 4 confirm this trend, with DurMI maintaining superior detection rates across a wide range of false positive rates. Importantly, DurMI is the only method that achieves strong performance on Wave-Grad2, while existing methods perform near random guessing as shown in Table 5. This highlights the advantage of operating on alignment signals upstream of the decoder, rather than relying on modality-sensitive denoising losses. As shown in Table 6, DurMI achieves high AUC on VITS2, but its TPR@1% FPR is comparatively lower. This can be attributed to VITS2's stochastic duration modeling, which weakens attack precision. The effect is further amplified on multi-speaker datasets like LibriTTS and VCTK, where greater variability in speech patterns reduces detection accuracy at low FPR.

**Efficiency Comparison.** DurMI is significantly faster than all baseline methods. As shown in Table 7, it requires only a single forward pass through the duration predictor, bypassing the diffusion

Table 5: Performance of MIA methods on WaveGrad2 across various datasets.

| | LJSpeech | | LibriTTS | | VCTK | |
|---|---|---|---|---|---|---|
| | AUC | TPR@1% FPR | AUC | TPR@1% FPR | AUC | TPR@1% FPR |
| Naive Attack | 50.1 | 1.0 | 54.3 | 0.6 | 59.9 | 1.5 |
| SecMI | 49.4 | 1.0 | 47.6 | 0.3 | 55.4 | 1.0 |
| PIA | 50.8 | 0.4 | 51.7 | 0.1 | 52.1 | 0.8 |
| PIAN | 50.3 | 0.1 | 50.2 | 0.1 | 44.7 | 0.1 |
| DurMI | **99.9** | **100.0** | **100.0** | **100.0** | **97.4** | **47.0** |

Table 6: Performance of DurMI on different TTS models across various datasets.

| | LJSpeech | | LibriTTS | | VCTK | |
|---|---|---|---|---|---|---|
| Model | AUC | TPR@1% FPR | AUC | TPR@1% FPR | AUC | TPR@1% FPR |
| VoiceFlow | 99.2 | 93.9 | 98.0 | 56.5 | 98.9 | 90.6 |
| FastSpeech2 | 100.0 | 100.0 | 99.2 | 90.5 | 99.5 | 93.7 |
| VITS2 | 97.5 | 80.1 | 85.5 | 22.4 | 87.1 | 12.2 |

decoding process entirely. This makes DurMI approximately 100× faster than SecMI and 50× faster than PIA for per-sample inference.

**Alignment Mechanisms.** DurMI achieves better performance on WaveGrad2 than on Grad-TTS, which we attribute to differences in alignment. WaveGrad2 relies on MFA, a fixed aligner prone to overfitting, thereby amplifying membership signals. In contrast, Grad-TTS employs MAS, a differentiable and adaptive aligner that reduces sample-specific memorization and weakens attack effectiveness. Notably, DurMI also performs strongly on VITS2, which adopts stochastic duration modeling rather than deterministic alignment, indicating that DurMI generalizes to probabilistic alignment strategies as well.

**Dataset.** On VCTK, DurMI attains high AUC but relatively lower TPR@1%FPR compared to other datasets. We attribute this to dataset-specific characteristics, including shorter utterances and lower text overlap between training and test sets. Full analysis of speaker composition, utterance length, and vocabulary overlap is provided in Appendix A.6. Even when text overlap is intentionally minimized, the practical implications remain significant (see Appendix A.6.3).

### 5.1 ABLATION STUDY

We conduct ablation studies on Grad-TTS using the VCTK dataset to examine the impact of various factors.

**Training Epochs.** As shown in Figure 3, increasing the number of training epochs leads to overfitting and better MIA performance. The performance plateaus after 1,000 epochs, suggesting 1,000–2,000 epochs as a practical range.

**Distance Metric for Duration Loss.** Figure 3 shows that $L_2$-norm (MSE) provides the best results. It aligns more closely with MAS-derived targets, strengthening memorization and member/non-member separation.

**Sensitivity to Utterance Length.** Based on the top and bottom 10% of utterance lengths, we divided the data into two clusters and assessed each of the four possible combinations of member and non-member clusters. Across various configurations, DurMI continuously showed distinct separability, suggesting robustness to input length. Appendix A.5 contains detailed visualizations of separability across different utterance durations.

Table 7: Running time (in milliseconds) for performing MIA on a single sample.

| | Inference Time (ms) | | | | |
|---|---|---|---|---|---|
| | Naive | SecMI | PIA | PIAN | DurMI |
| GradTTS | 1.54 | 3.04 | 1.53 | 1.53 | **0.03** |
| WaveGrad2 | 1.83 | 3.84 | 1.94 | 1.79 | **0.04** |

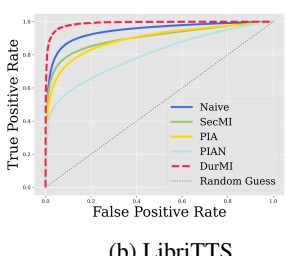

(h) Effect of training epochs.

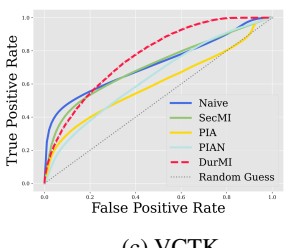

(i) Effect of distance metrics.

Figure 3: Ablation study of DurMI.

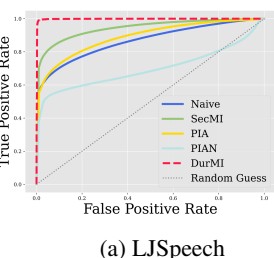

(a) LJSpeech

(b) LibriTTS

(c) VCTK

Figure 4: ROC curves comparing MIA methods on the Grad-TTS model across various datasets.

**Dataset Volume/Sampling Rate.** We examine how DurMI performance is affected by changes in the number of training and evaluation samples. Table 22 summarizes the effects of reducing the size of training and evaluation data. Our results show that DurMI is stable and robust even against limited sample provision. These show that the observed tendencies persist despite decreased data availability. Table 21 also shows that DurMI consistently maintains optimal performance at the default 22 kHz sampling rate. Detailed analysis is provided in Appendix A.13and A.12.

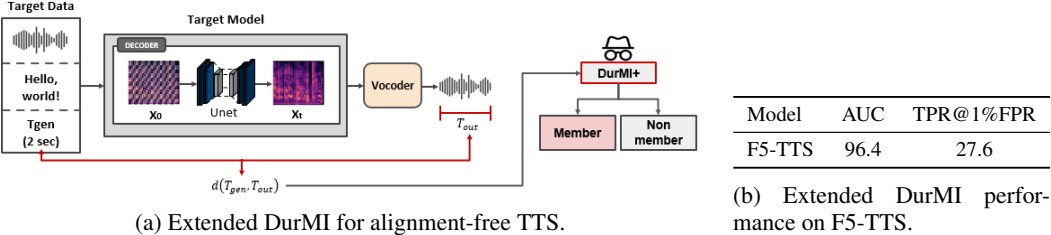

(a) Extended DurMI for alignment-free TTS.

| Model | AUC | TPR@1%FPR |
|---|---|---|
| F5-TTS | 96.4 | 27.6 |

(b) Extended DurMI performance on F5-TTS.

Figure 5: Extended DurMI overview and performance on F5-TTS.

# 6 EXTENDING DURMI TO ALIGNMENT-FREE AND ZERO-SHOT TTS

To account for recent trends in alignment-free TTS, we evaluate an extended version of DurMI on F5-TTS using a large-scale pretrained checkpoint under mismatched training and testing distributions (Emilia as the member dataset and LJSpeech as the non-member dataset) with prompt-based generation. Unlike prior work that relies on splitting a single dataset into member and non-member partitions, thereby preserving the same underlying distribution, our setting evaluates completely distinct datasets, providing a more realistic approximation of real-world deployment conditions.

The extended DurMI presents a clear separation between member and non-member distributions, indicating its applicability to alignment-free and zero-shot TTS systems. Empirically, member utterances consistently exhibit larger length deviations, that is, $d(T_{\mathrm{gen}}, T_{\mathrm{out}})$ is higher for training samples. This reversed trend arises from the characteristics of infilling-based TTS models. During inference, the model receives the original audio length $T_{\mathrm{gen}}$ as a control signal and produces an output with duration $T_{\mathrm{out}}$. Because infilling architectures learn alignment implicitly (e.g., via reconstruction of masked regions rather than explicit duration supervision), the model becomes tightly entangled with the fine-grained prosodic and timing patterns of its training recordings. As a result, it tends to overfit

to idiosyncratic duration irregularities present in the training utterances. When reconstructing these samples, the unstable implicit alignment mechanism induces greater variability in duration prediction. In contrast, unseen non-member utterances rely more on the model's global duration prior, yielding more stable and conservative predictions. Consequently, duration discrepancy emerges as a valid membership signal not because training examples match the target more accurately, but because they trigger stronger overfitting-dependent variability in duration reconstruction.

Given this tendency, the following duration discrepancy, denoted using $N_g$ ($N_{gen}$) and $N_o$ ($N_{out}$), can be computed using a variety of distance functions:

$$d(N_{\mathrm{g}}, N_{\mathrm{o}}) \in \{ |N_{\mathrm{g}} - N_{\mathrm{o}}|, \ (N_{\mathrm{g}} - N_{\mathrm{o}})^2, \ \tfrac{|N_{\mathrm{g}} - N_{\mathrm{o}}|}{N_{\mathrm{g}}}, \mathrm{Huber}_\delta(N_{\mathrm{g}} - N_{\mathrm{o}}), \ \mathrm{KL}(P_{N_{\mathrm{g}}} \| P_{N_{\mathrm{o}}}) \} \quad (7)$$

each capturing a distinct element of mismatch: distributional divergence, resilience via Huber loss, raw deviation, large-error sensitivity, and normalization across different utterance durations. Because of its flexibility, this indicator leads to a broad range of adaptation across different evaluation needs and dataset features.

The extended DurMI achieves a high AUC with a TPR of 28%, as demonstrated in Table 5b. Although its TPR is lower than that of explicit alignment-based models, it still demonstrates a meaningful level of membership signal detection. The extended DurMI design is visualized in Figure 5a.

These observations collectively underscore that alignment signals—whether explicit or implicit—play a central role in privacy leakage for modern TTS models. While DurMI cannot be directly applied to fully zero-shot models, proxy indicators such as $d(T_{\mathrm{gen}}, T_{\mathrm{out}})$ offer a principled path to extend our methodology in alignment-free settings. Importantly, this proxy requires only black-box access, making it feasible for real-world deployments. Additional candidate indicators for alignment-free TTS are provided in Appendix A.7.

## 7 DISCUSSION

**Defenses.** We apply DP-SGD to the duration predictor in Grad-TTS, controlling privacy via noise multipliers ($\sigma$) that determine the budget ($\varepsilon$) (details in Appendix A.8). As shown in Table 14, DP-SGD increases early training loss and reduces overfitting, yet DurMI retains strong attack performance across ($\varepsilon, \sigma$) settings; even at $\varepsilon \approx 1.6$, AUROC remains 98.7% and TPR@1%FPR stays at 83.3%. Because the duration module continues to leak membership information even with $\varepsilon \leq 10$, DP-SGD alone is insufficient, motivating the need for more robust and TTS-specific defense mechanisms.

**MIA Classifiers.** Attack performance degrades on stochastic-duration architectures because probabilistic duration sampling introduces high variance in output lengths, weakening the consistency of duration-loss signals and reducing member–non-member separability. Motivated by this limitation, we further evaluated a deep learning–based MIA classifier on GradTTS, which outperformed simple threshold-based attacks with higher TPRs across datasets(Table 15). A detailed description of these classifiers and the corresponding empirical analysis is provided in Appendix A.9.

**Train-attack Aligner Mismatch.** We evaluate DurMI under realistic conditions where the attacker's aligner differs from the aligner used during training, considering all MAS–MFA combinations. As shown in Table 20 of Appendix A.11, mismatched aligners significantly reduce membership inference success, highlighting the sensitivity of DurMI to alignment consistency. Nevertheless, matched-aligner settings remain practical in real-world scenarios because TTS pipelines often reveal or allow inference of the aligner used during training.

## 8 CONCLUSION

We propose DurMI, a grey-box membership inference attack that leverages duration loss to achieve higher accuracy and lower computational cost than prior approaches. Experiments across five TTS architectures show that duration supervision carries strong sample-specific signals, exposing a critical vulnerability in modern TTS pipelines.

## 9 ETHICS STATEMENT

This work adheres to the ICLR Code of Ethics. No studies involving human subjects or sensitive data were conducted beyond standard publicly available datasets. Potential ethical considerations, including fairness, privacy, and research integrity, have been carefully evaluated, and no conflicts of interest or harmful outcomes are expected.

## 10 REPRODUCIBILITY STATEMENT

The complete codebase for our experiments is organized into three primary components: (1) `attack/`, containing model-specific MIA implementations; (2) `train/`, which includes training scripts for all TTS models; and (3) `README` files, offering step-by-step instructions for data preprocessing, model training, and evaluation.

We release all pretrained model checkpoints and the corresponding preprocessed datasets at `https://zenodo.org/records/15474571`. The release includes all model-dataset combinations (Grad-TTS, WaveGrad2, and VoiceFlow across LJSpeech, LibriTTS, and VCTK), totaling nine checkpoints. All MIA methods can be directly evaluated using the provided resources.

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

## A APPENDIX

### A.1 EXPERIMENTAL SETUP

We trained two diffusion-based TTS models (Grad-TTS and WaveGrad2), one transformer-based model (FastSpeech2), one flow-matching-based model (VoiceFlow), and a stochastic duration model (VITS2) on three benchmark datasets: LJSpeech Ito & Johnson (2017), LibriTTS Zen et al. (2019), and VCTK Yamagishi et al. (2019). LJSpeech is a single-speaker dataset containing approximately 13,000 short utterances. In contrast, VCTK consists of recordings from 110 English speakers totaling around 43,000 samples, while a 20,000-sample subset of LibriTTS – a large-scale multi-speaker corpus – was used in our experiments.

Each dataset was split into two disjoint subsets: one for training (member set) and the other for evaluation (non-member set). All models were trained using a batch size of 16, a learning rate of

$1 \cdot e^{-4}$, and diffusion time steps – 50 for Grad-TTS and 1000 for WaveGrad2. All experiments were conducted on a single NVIDIA RTX A6000 GPU (48 GB VRAM), using the original model hyperparameters described below.

**Grad-TTS.** Grad-TTS was trained using a 22.05 kHz sampling rate and 80-dimensional mel-spectrograms, with an Fast Fourier Transform (FFT) size of 1024 and a hop length of 256. The encoder architecture comprises six convolutional layers (kernel size of 3, 192 channels), followed by two-headed multi-head attention and a dropout rate of 0.1.

**WaveGrad2.** WaveGrad2 was trained using a 22.05 kHz sampling rate and a hop length of 300. Its encoder consists of three convolutional layers with a kernel size of 5, 512 channels, and a dropout rate of 0.5.

**VoiceFlow.** VoiceFlow was trained on the same paired text-audio datasets as Grad-TTS and Wave-Grad2, using a 16 kHz sampling rate and 80-dimensional mel-spectrograms. The model was trained for 3,000 epochs with a batch size of 10 and a learning rate of $5 \cdot 10^{-5}$. Its encoder consists of six layers, each with 192 channels, a kernel size of 3, a dropout rate of 0.1, and two-headed multi-head attention. The filter channel size was set to 768, and the hop length to 200.

**FastSpeech2.** FastSpeech2 adopts a transformer-based architecture with four encoder layers and six decoder layers, each with a hidden size of 256 and two attention heads. The feed-forward network employs a filter size of 1024 with convolutional kernels of size [9, 1], while both encoder and decoder apply a dropout rate of 0.2. Variance predictors for pitch and energy use a filter size of 256, kernel size of 3, and a dropout rate of 0.5, with linear quantization into 256 bins.

**VITS2.** VITS2 was trained with a sampling rate of 22.05 kHz, using 80-dimensional mel-spectrograms (FFT size 2048, hop length 256, window length 1024). The model integrates variational inference with stochastic duration modeling, employing a transformer-based text encoder with six layers, two attention heads, and hidden dimensionality of 192. The decoder incorporates eight normalizing flows and a HiFi-GAN–style generator with multi-scale residual blocks (kernel sizes 3, 7, 11). Notably, unlike deterministic duration models, VITS2 uses stochastic duration prediction, enabling more diverse prosody modeling.

## A.2 BASELINE MIA CONFIGURATIONS

Grad-TTS and VoiceFlow are continuous-time diffusion models, where the diffusion timestep $t$ is sampled from the interval $[0, 1]$. For all attack methods – Naive Attack, SecMI, and PIA – we fix the number of diffusion timesteps to 100. Following their original implementations, both the Naive Attack and SecMI compute sample-wise reconstruction errors using the $\ell_2$ norm between the predicted and ground-truth noise at each timestep. In contrast, PIA adopts the $\ell_4$ norm to place greater emphasis on large errors, thereby increasing sensitivity to outliers in the denoising process.

In contrast, WaveGrad2 is a discrete-time diffusion model. According to the original codebase, the Naive Attack is performed with 100 discrete timesteps. For SecMI and PIA, the diffusion process is run with 1,000 timesteps, from which 100 are uniformly sampled at intervals of 10 to reduce computational overhead. The same norm configurations are applied: $\ell_2$ for Naive and SecMI, and $\ell_4$ for PIA.

## A.3 TEXT AND AUDIO DATA PREPROCESSING AND ALIGNMENTS

The preprocessing stage in TTS models involves processing both the encoder inputs (text) and decoder inputs (audio), as well as computing phoneme-to-audio alignments through an aligner to estimate target durations. Below, we describe the preprocessing and alignment procedures adopted for Grad-TTS, WaveGrad2, and VoiceFlow.

Grad-TTS Grad-TTS does not require explicit text normalization, and the audio can be used at its native sampling rate without resampling: 22,050 Hz for LJSpeech, 16,000 Hz for LibriTTS, and 48,000 Hz for VCTK. For alignment, Grad-TTS employs MAS to estimate target durations, implemented through the `monotonic_align` module compiled with Cython.

Table 8: Performance of DurMI across datasets and models.

| Model | Dataset | AUC | TPR@1% FPR |
|---|---|---|---|
| GradTTS | LJSpeech | $99.8 \pm 0.0$ | $98.83 \pm 0.06$ |
| | LibriTTS | $98.9 \pm 0.0$ | $83.5 \pm 0.0$ |
| | VCTK | $76.8 \pm 0.0$ | $9.6 \pm 0.0$ |
| WaveGrad2 | LJSpeech | $99.9 \pm 0.0$ | $100.0 \pm 0.0$ |
| | LibriTTS | $100.0 \pm 0.0$ | $100.0 \pm 0.0$ |
| | VCTK | $97.4 \pm 0.0$ | $50.97 \pm 0.06$ |

WaveGrad2 Text inputs are normalized by lowercasing and removing punctuation. All audio wave-forms are resampled to a consistent sampling rate of 22,050 Hz. Phoneme-to-audio alignments are generated using the MFA, which outputs alignment data in the TextGrid format. A TextGrid file includes tiered time-aligned annotations (e.g., phoneme and word levels), specifying the start and end time of each phoneme within the audio. These alignments are used to extract precise phoneme durations for training. Pre-generated TextGrid files for all datasets are provided and can be accessed at the following link: `https://drive.google.com/drive/folders/10eUTzOU06gTRMiQPoyw-Yctflms3ZLTJ?usp=sharing`.

**VoiceFlow**   To train the VoiceFlow model, the dataset must be organized in the Kaldi-style format. Accordingly, the preprocessing pipeline consists of two main stages: (1) metadata generation and (2) audio feature extraction. The following manifest files are created to structure and describe the dataset:

- `wav.scp`: It maps each utterance ID (typically the filename) to its corresponding audio file path.

- `utts.list`: It lists all utterance IDs extracted from `wav.scp`.

- `utt2spk`: It associates each utterance ID with a speaker ID. For single-speaker datasets like LJSpeech, the same speaker ID is used for all utterances.

- `text`: It contains pairs of utterance IDs and their corresponding transcript texts.

- `phn_duration`: It provides phoneme-level alignments, specifying the start time and duration of each phoneme within an utterance. These are extracted from TextGrid files generated by the MFA.

After metadata creation, mel-spectrogram features are extracted from the audio data using Voice-Flow's feature extractor. The features are stored in the following Kaldi-compatible formats:

- `feats.ark`: A binary file containing the actual mel-spectrogram feature matrices.

- `feats.scp`: A text file mapping each utterance ID to the corresponding entry in the `.ark` file.

VoiceFlow supports phoneme alignment using either MAS or MFA. During training, it uses phoneme durations generated by MAS, rather than the precomputed durations from `phn_duration`. Finally, the phoneme-level transcripts are aligned to phone IDs listed in `phones.txt`, which are used as input to the model.

### A.4 STATISTICAL SIGNIFICANCE ANALYSIS

To ensure the reliability and robustness of our findings, we conducted each DurMI experiment three times across both Grad-TTS and WaveGrad2 models, using all three datasets: LJSpeech, LibriTTS, and VCTK. The aggregated results, including mean values and standard deviations, are presented in Table 8. All iterations produced consistent outcomes, with negligible standard deviations (less than 0.1%).

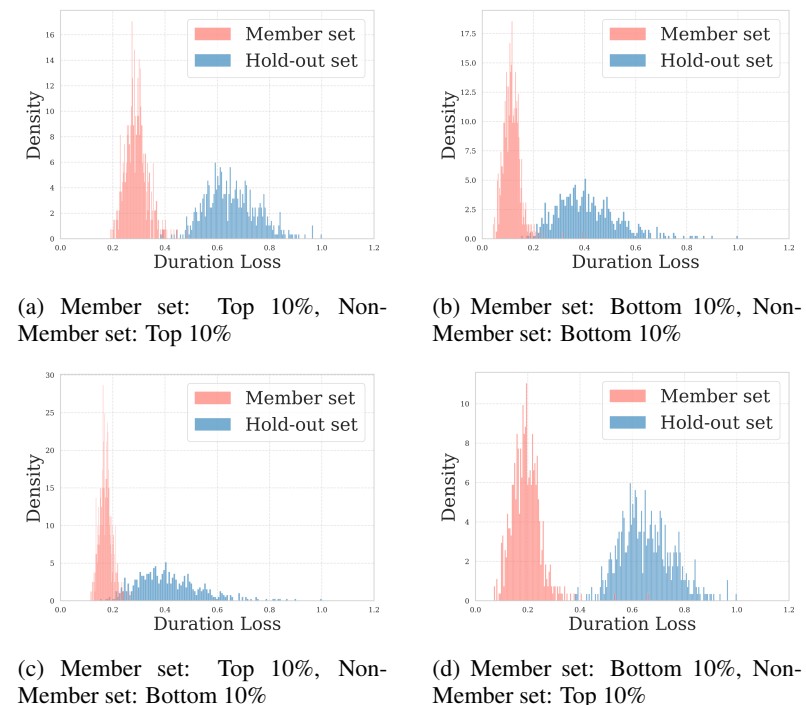

(a) Member set: Top 10%, Non-Member set: Top 10%

(b) Member set: Bottom 10%, Non-Member set: Bottom 10%

(c) Member set: Top 10%, Non-Member set: Bottom 10%

(d) Member set: Bottom 10%, Non-Member set: Top 10%

Figure 6: Performance comparison of DurMI across different utterance length clusters.

### A.5 ABLATION STUDY: IMPACT OF UTTERANCE LENGTH

TTS models typically incorporate a phoneme-level duration predictor, trained to minimize the discrepancy between predicted and actual phoneme durations. Longer utterances, which contain more phonemes, are prone to cumulative prediction errors. In addition, variations in pronunciation and prosody increase the modeling complexity for longer utterances. If such utterances are underrepresented in the training data, the model's generalization ability to these cases may be further limited.

Based on these considerations, we hypothesize that utterance length could influence the success rate of membership inference attacks (MIA). To investigate this, we conducted an ablation analysis using the Grad-TTS model and the LJSpeech dataset. We divided the dataset into clusters based on utterance length, selecting the top and bottom 10% of utterances. We then evaluated MIA performance across the following four cluster combinations:

1. Both member and non-member samples belong to the top 10% of utterance lengths (Cluster 1).

2. Both member and non-member samples belong to the bottom 10% (Cluster 2).

3. Member samples are from the top 10%, and non-member samples from the bottom 10% (Cluster 3).

4. Member samples are from the bottom 10%, and non-member samples from the top 10% (Cluster 4).

Figure 6 shows the distribution of duration loss across the four clusters. In all cases, member and non-member samples remain clearly separable.

Table 9 reports the AUC and TPR@1%FPR for each cluster. Although Cluster 3 shows a modest decrease (approximately 5 percentage points) in TPR@1%FPR, the overall impact of utterance length on DurMI performance is minimal.

In summary, the experimental results (Figure 6 and Table 9) indicate that utterance length does not significantly influence the effectiveness of MIA. Despite the reliance on duration information, the

Table 9: Membership inference performance across different utterance length conditions. Each subset contains approximately 600 samples.

| Condition | AUC | TPR @1% FPR |
|---|---|---|
| Full dataset | 99.8 | 98.9 |
| Cluster 1 | 99.9 | 98.9 |
| Cluster 2 | 99.8 | 98.9 |
| Cluster 3 | 99.4 | 94.9 |
| Cluster 4 | 99.8 | 100.0 |

Table 10: GradTTS-VCTK AUROC and FPR@1%TPR under different overlap conditions.

| | Speaker Overlap Present | Speaker Overlap Absent |
|---|---|---|
| Text Overlap Present | 96.8 / 9.6 | 82.25 / 10.47 |
| Text Overlap Absent | 85.90 / 22.04 | 85.91 / 22.27 |

variation in utterance length yields only marginal changes in duration loss, suggesting limited impact on attack success.

## A.6 DATASET ANALYSIS AND OVERLAP EFFECTS

To contextualize the results reported in Tables 4 and 6, we provide additional analysis of dataset-specific factors, focusing on speaker/text overlap and corpus statistics. This supplementary analysis clarifies why certain datasets, particularly VCTK, exhibit relatively lower TPR@1%FPR despite maintaining high AUROC.

### A.6.1 GRADTTS ON VCTK UNDER OVERLAP CONDITIONS

Table 10 reports GradTTS performance on VCTK when speaker and text overlaps are selectively allowed or removed.

The results show three key trends: (1) performance is strongest when both text and speaker overlap, (2) removing both types of overlap produces the lowest TPR, and (3) absence of text overlap degrades detection more severely than absence of speaker overlap. This indicates that text overlap is a decisive factor for generalization and detection accuracy.

### A.6.2 DATASET COMPOSITION

We also examine dataset-level statistics. Table 11 reports speaker splits, Table 12 shows average utterance length, and Table 13 summarizes vocabulary overlap.

- **Speakers:** VCTK includes 50 speakers with limited overlap, while LibriTTS has 185 speakers and LJSpeech only one.
- **Utterance length:** VCTK utterances average 3.2s, much shorter than LJSpeech (6.6s) or LibriTTS (12.7s), reducing temporal context.
- **Word overlap:** VCTK shows fewer shared words (4,694) compared to LJSpeech (8,631) and LibriTTS (12,633), limiting textual redundancy.

### A.6.3 TEXT OVERLAP AND REAL-WORLD IMPLICATIONS

For GradTTS–VCTK, AUROC remains above 85 even when all text overlap is removed, and TPR@1% FPR can even increase in these settings. This indicates that the attack signal does not primarily arise from repeated text prompts, but instead from other forms of overfitting, such as:

- speaker-specific acoustic and prosodic patterns,

Table 11: Speaker composition across datasets.

| Dataset | Total Speakers | Member | Non-member | Overlap |
|---------|---------------|--------|-----------|---------|
| LJSpeech | 1 | 1 | 1 | 1 |
| VCTK | 50 | 25 | 26 | 1 |
| LibriTTS | 185 | 89 | 87 | 1 |

Table 12: Average utterance length (sec).

| Dataset | Member | Non-member |
|---------|--------|-----------|
| LJSpeech | 6.56 | 6.59 |
| VCTK | 3.24 | 3.30 |
| LibriTTS | 12.67 | 12.67 |

- utterance-length regularities, and

- pronunciation and articulation characteristics

Therefore, minimizing text overlap alone could not be an effective defense strategy. In practice, membership leakage can still occur whenever the training data includes the same speakers or similar recording conditions, even if the textual content is entirely disjoint.

More robust mitigation strategies should focus on:

- increasing speaker diversity,

- limiting per-speaker data, and

- applying normalization or noise-based augmentation.

These observations suggest that real-world deployments remain vulnerable even under intentionally minimized text overlap, unless broader data- and model-level defenses are employed.

### A.6.4 REMARKS

These analyses explain why VCTK yields lower TPR@1%FPR despite high AUROC. Short utterances and reduced word overlap limit both temporal and lexical cues, making membership detection harder. By contrast, datasets with greater redundancy (LJSpeech, LibriTTS) provide stronger signals. Overall, dataset composition—particularly text overlap—plays a central role in shaping membership inference performance.

### A.7 PROXY INDICATORS FOR IMPLICIT DURATION MODELS

Recent alignment-free or zero-shot text-to-speech (TTS) models, including E2-TTS, F5-TTS, Seed-TTS (Anastassiou et al., 2024), and MaskGCT (Wang et al., 2024), do not rely on explicit duration modules. Instead, they are trained using infilling objectives, where random audio segments are masked and reconstructed given the remaining context and full text. This procedure enables zero-shot synthesis across unseen text-speaker pairs, while implicitly learning alignment between text and audio sequences. Architecturally, models such as E2-TTS employ only a mel-spectrogram generator and a vocoder, without phoneme-level duration supervision.

Although DurMI cannot be directly applied to these systems, implicit alignment cues remain exploitable for membership inference. We introduce two proxy indicators and one is discussed in Section 6 and the other is detailed in this section.

Infilling-based TTS training requires reconstructing masked frames. For training data, reconstruction tends to be more accurate, preserving pronunciation, timing, and speaker traits. Non-member samples often exhibit lower fidelity. Thus, similarity between original and reconstructed segments provides another membership signal.

Table 13: Word overlap between member and non-member subsets.

| Dataset | Member-only | Non-member-only | Shared |
|---------|-------------|-----------------|--------|
| LJSpeech | 7,123 | 7,189 | 8,631 |
| VCTK | 1,624 | 1,410 | 4,694 |
| LibriTTS | 8,221 | 7,789 | 12,633 |

## A.8 DEFENSES

We apply DP-SGD, one of widely used MIA defenses, to the duration predictor in Grad-TTS as our defense mechanism. DP-SGD reduces the influence of individual training samples by clipping gradients and injecting Gaussian noise. The strength of differential privacy is controlled by the privacy budget $\varepsilon$, which corresponds to different Gaussian noise multipliers $\sigma$. This defense has been shown to preserve practical TTS model quality while mitigating the exposure of duration-loss signals.

We vary the noise multiplier $\sigma$ under a fixed $\delta = 10^{-5}$ to obtain different privacy budgets $\varepsilon$, and evaluate both the convergence behavior of the duration loss and the resulting membership inference attack (MIA) performance.

Table 14 reports the duration loss of Grad-TTS and the corresponding attack performance (AUROC, TPR@1%FPR) across different $\varepsilon$–$\sigma$ configurations. The $\varepsilon$ values shown in the first row are computed with $\delta = 10^{-5}$, and all duration loss values are measured at Epoch 1000.

Applying DP-SGD to the duration predictor increases early training loss and suppresses overfitting, which could in principle hinder membership inference. However, DurMI remains highly effective throughout training. Even under our strongest tested DP setting ($\varepsilon \approx 1.6$), the AUROC stays at 98.7% (only 0.9% lower than non-private), and TPR@1%FPR remains high at 83.3%.

Importantly, **our DP experiments show that $\varepsilon$ values up to $\varepsilon \approx 10$ yield only minimal degradation in both model utility and attack performance**. These observations indicate that although DP-SGD stabilizes training and reduces early memorization, the duration module in Grad-TTS intrinsically leaks membership information, and the attack remains strong even under substantial DP noise.

| **Metric** | **Non-private** | $\epsilon = 10.961$ $\sigma = 1.0$ | $\epsilon = 5.187$ $\sigma = 1.5759$ | $\epsilon = 1.624$ $\sigma = 4.0234$ | $\epsilon = 9.996$ $\sigma = 1.0492$ | $\epsilon = 0.993$ $\sigma = 6.25$ | $\epsilon = 0.093$ $\sigma = 60.0$ |
|------------|-----------------|-------------|-------------|-------------|-------------|-------------|-------------|
| Duration loss | 0.27 | 0.338 | 0.338 | 0.345 | 0.337 | 0.357 | 0.615 |
| AUROC | 99.6 | 98.7 | 98.6 | 98.7 | 98.7 | 98.6 | 93.4 |
| TPR@1%FPR | 95.2 | 83.6 | 83.0 | 83.3 | 84.2 | 80.8 | 36.0 |

Table 14: Effect of DP-SGD noise levels on Duration Loss and Membership Inference Attack Performance in Grad-TTS

## A.9 LIMITATIONS OF DURATION-BASED MIAS AND DEEP LEARNING–BASED MIA.

Attack performance decreases on stochastic-duration models (e.g., VITS2 on VCTK). This is not simply due to weaker duration-loss signals but rather to increased stochasticity in duration predictions, which makes the loss values dynamic and inconsistent across samples. As output length is probabilistically determined, duration loss varies even for identical text, weakening the membership signal.

This implies the following:

1. A duration-loss-based MIA becomes most reliable on deterministic-duration models.

2. It may become less effective for stochastic or non-deterministic architectures.

Our results show that duration-loss-based membership inference attacks have a significant drawback when used with stochastic-duration TTS designs. Stochastic methods, like VITS2, produce output lengths through stochastic sampling, in contrast to deterministic methods, where duration alignment

is constant and loss signals are steady. Because of this architecture, the observed duration loss is highly noisy and duration projections are inherently more variable. As a result, even when the same text is synthesized numerous times, the loss values fluctuate, lowering consistency and diminishing the separability between members and non-members. This explains the observed degradation in attack performance and suggests that stochasticity functions as a natural regularizer against duration-based MIAs.

These results indicate that the effectiveness of duration-loss–based attacks is highly dependent on architectural design. While deterministic systems appear vulnerable due to stable and predictable alignment behavior, stochastic-duration architectures obscure membership signals through randomness, making traditional threshold-based attacks insufficient. This highlights a broader implication: architectural decisions intended for perceptual quality or modeling flexibility may unintentionally contribute to privacy resilience.

Motivated by this limitation, we explored deep learning–based MIA classifiers on GradTTS. We use a Recursive Neural Network (RNN) and a Multilayer Perceptron (MLP) to classify the duration loss of member set and hold-out set. The RNN Classifier's architecture is composed of one layer of GRU, one fully connected layer, and a sigmoid layer. The GRU layer has 128 hidden dimension. The MLP Classifier consists of three hidden-ReLU layers and a final layer with sigmoid activation. This model outperformed our previous MIA methodology, which relied on simple threshold-based classification, showing higher TPRs across multiple datasets. The results are presented in Table 15.

In terms of TPRs, the RNN classifier performs comparably or somewhat better than thresholding on LJSpeech and LibriTTS. When applying the RNN classifier rather than thresholding on VCTK, the AUROC marginally drops, but the TPR increases, suggesting more reliable detection of membership signals across different thresholds. In general, membership inference across datasets becomes more reliable when applying an deep-learning-based attack model such as RNN, as opposed to straightforward thresholding.

| Dataset | Thresholding (prior) | RNN Classifier (new) |
|---------|---------------------|---------------------|
| LJSpeech | 99.7 / 99.1 | 99.1 / 98.8 |
| LibriTTS | 98.9 / 82.8 | 99.3 / 88.2 |
| VCTK | 86.7 / 18.2 | 73.8 / 26.7 |

Table 15: Comparison of MIA performance using thresholding vs. RNN-based classifier across different datasets.

### A.10    ADDITIONAL TPR RESULTS AT LOWER FPRS

We explored lower FPRs and computed the corresponding TPRs. Table 16 reports the results for Durmi, while Table 17, Table 18, and Table 19 present the results for our baseline models under the same settings.

### A.11    TRAIN–ATTACK ALIGNER MISMATCH

To evaluate the robustness of DurMI under realistic attack conditions, we analyze performance when the aligner used in the attack phase differs from the aligner used during model training. Specifically, we consider combinations of MAS and MFA aligners for both training and attack pipelines, reflecting a case where the attacker does not know the exact training configuration.

These results in Table 20 demonstrate that mismatched aligners substantially reduce membership inference success. However, note that matched-aligner settings can be practical in real-world scenarios because TTS pipelines typically disclose aligner choices (e.g., MFA, MAS, or proprietary tools), or they can be inferred from training scripts or released checkpoints.

### A.12    ABLATION STUDY: SAMPLING RATE

We conduct an additional sampling-rate ablation on GradTTS trained with VCTK to evaluate the sensitivity of DurMI to the sampling resolution of generated speech. As summarized in Table 21, membership inference performance peaks at **22 kHz**, which matches the model's default generation

Table 16: DurMI: TPR at different FPR thresholds

| Model | Dataset | TPR@1%FPR | TPR@0.1%FPR | TPR@0.01%FPR |
|---|---|---|---|---|
| **GradTTS** | LJSpeech | 99.1 | 36.9 | 7.1 |
| | LibriTTS | 82.8 | 50.4 | 0.2 |
| | VCTK 10k | 18.2 | 4.7 | 2.5 |
| **WaveGrad2** | LJSpeech | 100.0 | 100.0 | 99.9 |
| | LibriTTS | 100.0 | 100.0 | 100.0 |
| | VCTK | 47.0 | 7.9 | 0.5 |
| **FastSpeech2** | LJSpeech | 100.0 | 100.0 | 100.0 |
| | LibriTTS | 90.5 | 47.5 | 17.2 |
| | VCTK | 93.7 | 73.6 | 48.7 |
| **VoiceFlow** | LJSpeech | 93.9 | 63.9 | 50.7 |
| | LibriTTS | 56.5 | 18.6 | 2.1 |
| | VCTK | 90.6 | 73.5 | 47.1 |
| **VITS2** | LJSpeech | 80.1 | 42.7 | 20.7 |
| | LibriTTS | 22.4 | 5.4 | 5.4 |
| | VCTK | 12.2 | 0.1 | 0.1 |

Table 17: Naive Attack: TPR at different FPR thresholds

| Model | Dataset | TPR@1%FPR | TPR@0.1%FPR | TPR@0.01%FPR |
|---|---|---|---|---|
| **GradTTS** | LJSpeech | 55.0 | 19.7 | 3.6 |
| | LibriTTS | 58.1 | 33.7 | 12.8 |
| | VCTK | 29.5 | 5.7 | 3.9 |
| **WaveGrad2** | LJSpeech | 1.0 | 0.1 | 0.03 |
| | LibriTTS | 0.6 | 0.7 | 0.1 |
| | VCTK | 1.5 | 0.1 | 0.0 |

Table 18: SecMI: TPR at different FPR thresholds

| Model | Dataset | TPR@1%FPR | TPR@0.1%FPR | TPR@0.01%FPR |
|---|---|---|---|---|
| **GradTTS** | LJSpeech | 70.3 | 25.7 | 2.9 |
| | LibriTTS | 55.2 | 38.8 | 8.8 |
| | VCTK | 8.1 | 0.8 | 0.2 |
| **WaveGrad2** | LJSpeech | 1.0 | 0.1 | 0.02 |
| | LibriTTS | 0.3 | 0.1 | 0.02 |
| | VCTK | 1.0 | 0.5 | 0.1 |

Table 19: PIA: TPR at different FPR thresholds

| Model | Dataset | TPR@1%FPR | TPR@0.1%FPR | TPR@0.01%FPR |
|---|---|---|---|---|
| **GradTTS** | LJSpeech | 55.0 | 20.5 | 1.8 |
| | LibriTTS | 47.0 | 23.9 | 2.7 |
| | VCTK | 9.7 | 1.2 | 0.2 |
| **WaveGrad2** | LJSpeech | 0.4 | 0.1 | 0.06 |
| | LibriTTS | 0.1 | 0.06 | 0.04 |
| | VCTK | 0.8 | 0.7 | 0.3 |

Table 20: DurMI performance under training–attack aligner mismatch (VoiceFlow)

| Training → Attack | AUROC | TPR@1%FPR |
|---|---|---|
| MAS → MAS (matched) | **99.2** | **93.9** |
| MAS → MFA (mismatched) | 91.7 | 0.8 |
| MFA → MFA (matched) | **93.8** | **47.8** |
| MFA → MAS (mismatched) | 44.1 | 0.9 |

setting. Both excessively low and excessively high sampling rates degrade attack performance, indicating that the default 22 kHz configuration preserves the signal-level characteristics most relevant to DurMI.

Table 21: Ablation Study: Impact of Sampling-rate on GradTTS (VCTK)

| Sampling Rate (Hz) | AUROC | TPR@1%FPR |
|---|---|---|
| 100 | 38.9 | 0.2 |
| 1,000 | 39.3 | 0.2 |
| 10,000 | 48.5 | 0.3 |
| **22,050** | **76.8** | **9.6** |
| 30,000 | 75.0 | 4.7 |
| 40,000 | 62.4 | 1.0 |
| 220,500 | 40.7 | 3.0 |

### A.13 ABLATION STUDY: IMPACT OF DATASET VOLUME

We investigate the sensitivity of DurMI performance to variation in both training and evaluation sample counts. Table 22 summarizes the effect of reducing available data. Our results show that DurMI remains **stable and robust** even under substantial sample reductions.

Table 22: Effect of sample-size reduction on DurMI performance

| Model / Dataset | Original Setting | Reduced Samples | AUC Change | TPR@1%FPR Change |
|---|---|---|---|---|
| **GradTTS – LJSpeech** (Eval size: 10,000 → 1,000) | 99.7 / 99.1 | 99.6 / 95.2 | −0.1 | −3.9 |
| **GradTTS – VCTK** (Train size: full → 7,000) | 86.7 / 18.2 | 85.9 / 22.2 | −0.8 | +4.0 |

These results indicate that the observed trends are maintained despite reduced data availability, supporting the robustness and reproducibility of our findings.

### A.14 IMPORTANCE OF ALIGNMENT IN CONTEMPORARY TTS MODELS

Recent developments in text-to-speech (TTS) systems indicate that incorporating text–speech alignment information continues to play a critical role in generating accurate and stable speech, regardless of model architecture. Although Seed-TTS (Anastassiou et al., 2024) adopts an autoregressive framework and claims an alignment-free generation strategy, its design still relies on predicting the duration of the target utterance and leveraging this prediction to locally optimize alignment during generation. This suggests that implicit alignment mechanisms substantially contribute to reliable speech synthesis.

Similarly, MegaTTS-3 (Jiang et al., 2025) demonstrates that introducing sparse alignment supervision leads to significant improvements in robustness and intelligibility, particularly for long or syntactically complex sentences. In contrast, systems that omit explicit alignment often exhibit increased word skipping, incorrect timing, or unstable prosody when handling challenging input text.

Related observations are also reported in recent T5-TTS (Chen et al., 2024), where insufficient alignment often results in hallucinated phrases, repetitions, and semantic corruption. The application of monotonic alignment constraints has been shown to markedly reduce character-level error rates, indicating that alignment is essential not only for acoustic quality but also for semantic fidelity.

Taken together, these findings collectively suggest that despite the recent shift toward autoregressive, diffusion-based, or codec-LLM architectures, alignment—whether explicit, sparse, or implicitly encoded—remains a crucial component for high-quality and robust TTS, and it is unlikely to be eliminated from future system designs.

