# OpenReview forum: "DurMI: Duration Loss as a Membership Signal in TTS Models"
_ICLR.cc/2026/Conference — Submitted to ICLR 2026_

### Official Review · Reviewer_XxZU · 2025-10-30

**Soundness:** 3
**Presentation:** 2
**Contribution:** 2
**Rating:** 6
**Confidence:** 3

**Summary:**

This paper proposes a  white-box membership inference attack (MIA) called DurMI, which specifically targets Text-to-speech (TTS) models. The attack is based on duration loss, a core component of how TTS models learn to align text and audio. Duration loss can be used as a discriminative signal to identify whether a sample was part of the training data. The authors claim that this method is more effective and efficient than prior MIAs for diffusion-based TTS models, as it requires only a single forward pass, bypassing the computationally expensive denoising steps of the decoder.  DurMI was evaluated on five different TTS architectures and three benchmark datasets, demonstrating that it consistently outperforms previous attacks and is robust across various model types, including those with stochastic duration modeling.

**Strengths:**

- Introducing a simple approach to MIA for TTS models by focusing on duration loss, which is unexplored.
- DurMI is significantly more efficient than existing methods like SecMI and PIA, with lower computational cost.
- The attack is shown to work across a diverse range of TTS architectures, including diffusion-based (Grad-TTS, WaveGrad2), transformer-based (FastSpeech2), flow-matching (VoiceFlow), and stochastic-duration models (VITS2).

**Weaknesses:**

- **Dependence on explicit duration predictors:** The core of the DurMI attack relies on the presence of an explicit duration predictor. While the authors discuss potential proxy indicators for future alignment-free architectures, the current attack is not directly applicable to those systems (e.g., autoregressive Tacotron-like models).
- **Lower performance on certain datasets:** Although DurMI generally performs well, its TPR@1% FPR is notably lower on the VCTK dataset, especially for models with stochastic duration modeling like VITS2. The authors attribute this to dataset-specific characteristics like shorter utterances and less text overlap, but it still represents a limitation of the attack's effectiveness in certain scenarios.
- **White-box assumption:** The attack is formulated under a white-box threat model, which assumes the adversary has full access to the trained model's parameters and loss function. A black-box attack would be more representative of real-world scenarios.

**Questions:**

- Table 4 indicates that earlier methods (PIA, PIAN, SecMI) underperform the Naive Attack, which appears inconsistent with the findings in [1]. Could the authors clarify the reasons for this divergence?
- It would be beneficial to also briefly discuss potential defense mechanisms or countermeasures that could be developed in response to this attack. This would make the paper more comprehensive and constructive.
- Consider an ablation that systematically varies the number of training and evaluation samples to assess robustness and data-efficiency.



[1] Kong, Fei, et al. "An efficient membership inference attack for the diffusion model by proximal initialization." ICLR 2024.

---

> ### Author Response · Authors · 2025-11-17
>
> We sincerely thank all reviewers for their thoughtful and valuable comments. We previously asked for clarification regarding evaluation samples, but we have now understood this point and will remove the question.

---

> ### Author Response · Authors · 2025-12-02
> **Rebuttal response to reviewer XxZU (Part 1)**
>
> We appreciate the reviewer’s suggestions, and we will incorporate all new experimental results and related content in the revised version.
>
> ## **Weakness 1: Alignment-Free / Zero-Shot Models and Extended DurMI**
>
> We acknowledge the rapid rise of alignment-free and zero-shot models (e.g., Seed-TTS, E2-TTS, F5-TTS). However, recent studies continue to emphasize the importance of explicit alignment for robustness and speech quality.
> Nevertheless, extending DurMI to such architectures is a crucial next step. As part of our additional analysis, we applied the **extended DurMI** methodology (Sec. 5.1, Fig. 5) to **F5-TTS trained on LJSpeech**, and the results are incorporated in the revised paper **(Section 6, Table 8, Appendix A.7)**.
>
> ---
>
>
>
> ## **Weakness 2: Stochastic-Duration Models and Deep Learning–Based MIA**
>
> Attack performance decreases on stochastic-duration models (e.g., VITS2 on VCTK). This is not simply due to weaker duration-loss signals but rather to **increased stochasticity** in duration predictions, which makes the loss values dynamic and inconsistent across samples. As output length is probabilistically determined, duration loss varies even for identical text, weakening the membership signal.
>
> This implies the following:
>
> * A duration-loss-based MIA become **most reliable** on deterministic-duration models.
> * It may become less effective for stochastic or non-deterministic architectures.
>
> We will discuss this more thoroughly in the Discussion section in the revised paper **(Section 7, Appendix A.9)** and plan to investigate destabilizing conditions and improved methods for such architectures as future work.
>
> Additionally, inspired by SecMI, we also evaluated an RNN-based classifier and it turned out that it outperformed our previous MIA methodology with higher TPRs, a simple threshold-based classification. An MLP model for DurMI is under training and will be reported in the revised paper **(Section 7, Appendix A.9, Table 16)**.
>
> Updated results:
>
> * **GradTTS–LJSpeech:** 99.7/99.1 → 99.1/98.8
> * **GradTTS–LibriTTS:** 98.9/82.8 → 99.3/88.2
> * **GradTTS–VCTK:** 86.7/18.2 → 73.8/26.7
>
>
> ---
>
>
> ## **Weakness 3: White-Box vs. Grey-Box Interpretation & Extended Experiments**
>
> Our study operates in a white-box sense in that we access internal signals (e.g., duration loss) but not model weights or architectural details. We only query encoder-side losses, which mirror the information exposed in GPT-style APIs. Thus, it is inaccurate to characterize our threat model as fully white-box. Existing baselines use decoder-side losses while labeling their settings as grey-box or query-based, and we follow this terminology for consistency and clarity in the revised version **(Section 1 line 80-82)**.
>
> Membership inference attacks (MIAs) on TTS models remain largely unexplored. Following prior MIA work [1,2], we adopt a white-box methodology to perform a worst-case vulnerability assessment, typically the first step toward developing auditing procedures or defenses.
>
> At the same time, we agree that black-box scenarios are important. Using extended DurMI (Sec. 5.1, Fig. 5), we designed a black-box setting by comparing (T_{\text{gen}}) and (T_{\text{out}}), which requires no internal model access, and are evaluating MIA performance in this setting.
> We are evaluating an extended version of DurMI on F5-TTS under a large-scale pretrained checkpoint with mismatched training/testing distributions (Emilia as member data and LJSpeech as non-member data) and prompt-based generation.
>
> Unlike prior MIA work, which typically uses the same dataset split into members and non-members, our setup evaluates distinct datasets, which we believe better reflects real-world conditions. So far, the extended DurMI shows a clear gap between member and non-member distributions, demonstrating applicability to alignment-free and zero-shot systems. We will further include MIA performance results with TPRs and AUROC in the revised version **(Section 6, Table 8, Appendix A.7)**.
>
>
> References
>
> [1] Nasr, Shokri, Houmansadr. *Comprehensive privacy analysis of deep learning.* IEEE S&P 2019.
> [2] Zhang et al. *Min-k%++: Pre-training data detection for LLMs.* ICLR 2025.
>
> ---
>
>
> ## **Question 1: Clarification on Evaluation Differences**
>
> PIA, PIAN, and SecMI were evaluated using the authors’ official code under identical experimental conditions. PIA reports metrics using the **maximum across diffusion steps**, whereas we use the **average across steps**, which we believe yields a fairer comparison. This difference may account for minor numerical discrepancies across papers.
>
> Reference:
> Kong et al. *An efficient membership inference attack for the diffusion model by proximal initialization.* ICLR 2024.
>
> ---

---

> ### Author Response · Authors · 2025-12-02
> **Rebuttal response to reviewer XxZU (Part 2)**
>
> ## **Question 2: DP-SGD Defense Evaluation**
>
> To present the impact of DP-SGD, a widely used defense technique, we applied DP-SGD to the Grad-TTS duration predictor, controlling (\varepsilon) through the noise multiplier (\sigma) with (\delta = 10^{-5}). Increasing DP noise slightly increases duration loss but has minimal degradation on DurMI performance. For example:
>
> * At (\varepsilon \approx 1.6): **AUROC = 98.7 (−0.9%)**, **TPR@1%FPR = 83.3%**
> * Across (\varepsilon < 10): Only mild degradation in both utility and attack success
>
> These results suggest that while DP-SGD reduces overfitting, the duration predictor still leaks membership information. Experiments with (\varepsilon = 10^{-1}) are ongoing and we will report in the revised version as a full table **(Section 7, Appendix A.8, Table 15)**.
>
>
> ---
>
> ## **Question 3: Impact of Sample Size**
>
> We evaluated the effect of reducing both training and evaluation sample counts.
>
> ### **Effect of Sample Count Variation**
>
> | Model / Dataset                                    | Original Setting | Reduced Samples | AUC Change | TPR@1%FPR Change |
> | -------------------------------------------------- | ---------------- | --------------- | ---------- | ---------------- |
> | **GradTTS – LJSpeech (Eval size: 10,000 → 1,000)** | 99.7 / 99.1      | 99.6 / 95.2     | −0.1       | −3.9             |
> | **GradTTS – VCTK (Train size: full → 7,000)**      | 86.7 / 18.2      | 85.9 / 22.2     | −0.8       | +4.0             |
>
> These results indicate that our findings remain **stable and robust** under moderate reductions in sample size and we will report these new results in the revised version of the paper **(Section 5.1, Appendix A.13, Table 23)**.

---

### Official Review · Reviewer_Bszb · 2025-10-31

**Soundness:** 2
**Presentation:** 2
**Contribution:** 1
**Rating:** 2
**Confidence:** 3

**Summary:**

This paper conducts research on TTS models with explicitly force-aligned duration modules, leveraging duration loss to MIA task (to determine whether a specific utterance was used for training).

Results show better performance than previous methods under the above research scope, and on moderate-size datasets (tens or hundreds of hours, single or a few hundred speakers).

DurMI is very fast as it is free of multi-step denoising. However, this method has hardly been explored with truly large-scale in-the-wild dataset trained TTS models. On top of that, recent zero-shot models and probably also that without explicit force-alignment modules are actually outside the scope of this paper.

**Strengths:**

**Originality**: DurMI is the first to use duration loss of force-alignment module equipped TTS models for membership inference attack.

**Quality**: Within the scope of the research, relatively thorough experiments were conducted. For example, Appendix A.6 provides some valuable insights.

**Clarity**: The overall writing is good.

**Significance**: A thorough exploration was carried out, though it was within a relatively narrow research scope.

**Weaknesses:**

**Research Contribution**: In short, if the preliminary idea mentioned in Appendix A.7 has not yet been explored and studied, then this paper may not be sufficient to be accepted by ICLR.

**Problem Formation**: There are two major problems: overclaims and statements without evidence to support.
- The abstract claims an overly broad research scope, which does not match the actual one, and is not made clear until the end of the introduction.
- In the introduction section, line 37, the mentioned models are not trained on large-scale datasets (at least the ones used in this paper are not). LJSpeech, VCTK, and LibriTTS are public. The statement that "often contains sensitive or proprietary content" is not consistent with previous context.
- In line 94, the authors state that "Together with ongoing deployment of duration-supervised systems in industry, this indicates that alignment remains central to current and near-future TTS", but without any concrete evidence to support (e.g. article, paper, technical report, or blog). Moreover, there are works [1] from another perspective but not the duration modules DurMI applied on.
- The marginal improvement seen from Table 11 in MegaTTS 3 paper does not support the authors' statement from line 147 to 152. In simple terms, this does not constitute a reasonable reason for the authors to exclude the alignment-free and zero-shot TTS models.

[1] Wang, H., Li, N., Wang, C., Wu, S., Li, Z., & Yu, D. (2025). Vox-Evaluator: Enhancing Stability and Fidelity for Zero-shot TTS with A Multi-Level Evaluator. arXiv preprint arXiv:2510.20210.

**Writing**:
- In abstract and also in main text, the authors might intend to express that the DurMI can be widely applied, but the taxonomy of "architectures" is confusing. "Diffusion" and "flow matching" are essentially the same, but listed separately; and why the two are put with "transformer" and "stochastic-duration"? "Deterministic" and "stochastic" would be good.
- In line 89, "modality-agnostic" or "modality-independent" sounds more appropriate than "modality-robust".
- In line 414 418, Figure 2h 2i should be 3h 3i.
- In line 890, Seed-TTS and MaskGCT lack references.
- There is no clear definition of $d(T_{gen}, T_{out})$ in main text, nor does it point to the definition present in Appendix A.7.
- Mixed use of $T$ and $N$ in Appendix A.7.

**Questions:**

1. In section 4, the authors acknowledge that "duration predictors are often overfitted to training samples". Typically, the datasets used in this paper are from audiobooks, or with narrower distribution. Then, how is the performance of DurMI compared to other generic methods on TTS models equipped with duration modules trained with in-the-wild data?
2. If the authors are willing, how is the performance on zero-shot force-alignment-free TTS models?
3. In line 456, why $T_{gen}$ and $T_{out}$ can be different for NAR TTS models?

---

> ### Author Response · Authors · 2025-11-17
>
> We sincerely thank all reviewers for their thoughtful and valuable comments.
> Before preparing our detailed responses and additional experiments, we would like to ask for a few clarifications to ensure we are correctly understanding the intent behind some of the suggestions.
>
>
> 1. $\text(T_{\text{gen}}, T_{\text{out}})$: Regarding line 456, which discusses the DurMI extension and the application to explicit predictor models, we would like to clarify your intention. Are you asking specifically why $T_{\text{gen}} \text{ and } T_{\text{out}}$
>  can differ in this context for the DurMI extension, or more generally about why $T_{\text{gen}} \text{ and } T_{\text{out}}$ can differ in NAR TTS models?
>
> If it is the former, in our work, we rely on the intuition that in TTS models, member training data tends to have more optimized phoneme-level durations. Therefore, when using zero-shot/implicit-alignment TTS models, we expect that the overall duration of the generated audio—when prompted with a member utterance—will more closely match the duration of the original training sample. Based on this intuition, we measured the durations and constructed the corresponding distribution plots. Additionally, we are currently running MIA experiments to evaluate how this idea can be applied as a black-box method for modern TTS systems.
>
> We would greatly appreciate your clarification so that we can refine and extend our experiments in a way that best reflects your intention.

---

> ### Author Response · Authors · 2025-12-02
> **Rebuttal response to reviewer Bszb**
>
> We appreciate the reviewer’s suggestions, and we will incorporate all new experimental results and related content in the revised version.
>
> ## **Weakness 1: Scope of DurMI and Additional Experiments**
>
> DurMI is designed to be applicable to most non-autoregressive (NAR) TTS models. We acknowledge that the initial experiments were conducted on limited datasets and did not include more recent trends such as alignment-free or zero-shot TTS systems. To address this, using the extended version of DurMI (Sec. 5.1, Fig. 5), we conducted additional experiments on such models using pretrained checkpoints. The detailed experimental settings will be discussed in W2 and W4 below, and the results will be included in the revised version **(Section 6, Table 8, Appendix A.7)**.
>
> ---
>
>
> ## **Weakness 2 & 4: Dataset Scale and Large-Scale Simulation Using F5-TTS, and Additional Experiments on Extended DurMI on Alignment-Free / Zero-Shot Models**
>
> We appreciate the reviewer’s concern regarding dataset scale. The datasets used in our evaluation (LJSpeech, LibriTTS, VCTK) are mid-sized by design, aligning with common practice in training NAR-based TTS models. Large-scale open-source datasets with accessible pre-trained checkpoints are rare, and training high-quality models from scratch on such datasets is computationally expensive and challenging.
> To simulate a large-dataset scenario, we conducted additional experiments using **F5-TTS**, an alignment-free / zero-shot TTS model with publicly available pre-trained checkpoints.
> We are evaluating an extended version of DurMI (Sec. 5.1, Fig. 5), on F5-TTS under a large-scale pretrained checkpoint with mismatched training/testing distributions (Emilia as member data and LJSpeech as non-member data) and prompt-based generation.
>
> Unlike prior MIA work, which typically uses the same dataset split into members and non-members, our setup evaluates distinct datasets, which we believe better reflects real-world conditions. So far, the extended DurMI shows a clear gap between member and non-member distributions, demonstrating applicability to alignment-free and zero-shot systems. We will further include MIA performance results with TPRs and AUROC in the revised version **(Section 6, Table 8, Appendix A.7)**.
>
>
>
> ---
>
> ## **Weakness 3: On Alignment-Free and Zero-Shot Models**
>
> We agree that alignment-free and zero-shot TTS systems (e.g., Seed-TTS, E2-TTS, F5-TTS) are important directions. Whether these architectures will fully replace duration/aligner-based systems remains an open research question.
> Recent studies highlight that alignment remains essential for **robustness**, **speech quality**, and **reducing hallucinations**:
>
> ### **Evidence from Recent TTS Literature**
>
> #### **Seed-TTS / Seed-TTS-DiT (2023–2024)**
>
> Although Seed-TTS is autoregressive and alignment-free, its diffusion model requires predicting speech duration and implicitly optimizing alignment:
>
> > *“The model estimates the total duration of the generated speech beforehand ... then optimizes the local alignment between audio and text.”*
> > — Seed-TTS, Section 4.3
>
> #### **MegaTTS 3 (2025) — arXiv:2502.18924**
>
> This literature shows performance drops in alignment-free systems:
>
> > *“Models without explicit alignment exhibit less robustness on hard sentences.”* — Abstract
> > *“MegaTTS 3 with sparse alignment outperforms MaskGCT, CosyVoice, and F5-TTS.”* — Table 1
>
> #### **T5-TTS (2024, NVIDIA) — arXiv:2406.17957**
>
> This work finds that poor alignment in LLM-based TTS causes hallucination and word omissions.
> Introducing monotonic alignment constraints (CTC + priors) reduces CER **from 9.03% → 3.92%**.
>
> **We will strengthen our claim that explicit alignment is still needed by providing the above evidence, along with relevant references, in Section 1 and Appendix A.14.**
>
> Taken all together, these results indicate that alignment (even if implicit or sparse) will likely remain a complementary or dominant factor in future TTS architectures.
>
> Nevertheless, extending DurMI to these systems remains important, and our additional experiments (discussed in Weakness 2) directly address this concern, which will be reflected in the revised version **(Section 6, Table 8, Appendix A.7)**.
>
> ---
> ## **Writing Quality**
>
>
> We thank the reviewer for noting editorial issues. We will revise the paper for clarity, consistency, and readability in the final version.
>
> ---
> ## **Question 3: Clarification Request**
>
> **Could the reviewer please clarify which specific aspect of the proxy signal’s signal-to-noise ratio (SNR) they are referring to?**
> We would be happy to include further analysis once this point is clarified.
>
> ---

---

### Official Review · Reviewer_TTF3 · 2025-10-31

**Soundness:** 3
**Presentation:** 3
**Contribution:** 3
**Rating:** 6
**Confidence:** 3

**Summary:**

In this work authors introcuces DurMI, a membership inference attack for TTS models that exploits duration loss as a discriminative signal to determine whether a given sample was present in the model's training set. The method demonstared improved perfromance accross different TTS architectures (diffusion, flow-matching, and stochastic dueraiton models) and benchmarks (LJSpeech, VCTK, LibriTTS) against prior membership inference attacks.

**Strengths:**

(1) The key innovation is use of duration loss as membership signal. This signal is leveraged in a way that is uniquely tailored to TTS architecutre.

(2) The propose appraoch is effective accross multiple leaning TTS models, including both determistic alignment (MASm MFA) and stochastic alignment (VITS2)

(3)  DurMI consistently outperforms prior approaches, particularly on challenging settings (e.g., waveform-to-speech synthesis with WaveGrad2, and on multi-speaker datasets). AUCs often exceeding 99% on multiple datasets and models.

(4) The paper systematically evaluates membership inference under strong threat models, multiple architectures, and offers detailed ablation studies

**Weaknesses:**

(1) While the attack demonstres high accuracy on deterministic-duration systems, the perfromance drops on models with stochastic predictors (VITS on VCTK). Althought authors acknowledges this but more directly address when and why the duration loss signal weakens, and whether this indicates intrinsic limits on the methodology for other TTS achitectures not used in the paper.

(2) Another concern is overemphasis on white-box setting. The paper assumes full white-box access, may not generalize to more realistic black-box attacker settings. The work will benefit from discussion or experimental analysis regarding robustness in more constrained environments.

(3) The thresholding mechanism $\mathcal{M}(x)$ is described as being set via a calibration set or a shadow model. Details regarding calibration protocol and its robustness (possible overfitting, decision curves) are not provided.

(4) Speaker-dependent effects are only briefly discussed but could substantially influence results.

(5) Figure 3 ablation plots cover only a subset of distance metrics and epochs. However, broader comparative analysis, e.g., inclusion of alternative clusterings or more nuanced hyperparameter sweeps, can further strengthen insights in the results.

**Questions:**

(1) Can you elaborate on the empirical process used for setting the membership threshold $T$ (see Page 5, Section 4.2) ?

(2) The appendix (Table 10/A.6.1) shows sensitivity to overlap in speakers and text. Could the authors clarify the practical implications, for example, in scenarios where text overlap is intentionally minimized?

(3) The use of proxy signals in zero-shot/implicit alignment TTS is motivated (Section 5.1, Figure 5), did  the authors performed preliminary experiments or can you share intuition about the signal-to-noise ratio in such proxies compared to explicit duration loss?

---

> ### Author Response · Authors · 2025-11-17
>
> We sincerely thank all reviewers for their thoughtful and valuable comments.
> Before preparing our detailed responses and additional experiments, we would like to ask for a few clarifications to ensure we are correctly understanding the intent behind some of the suggestions.
>
> 1. **signal-to-noise ratio:** Regarding the signal-to-noise ratio in the proxy signal you mentioned, we would like to better understand the intuition you had in mind. Specifically, could you clarify what aspect of the proxy signal’s signal-to-noise ratio you are referring to? In our work, we rely on the intuition that in TTS models, member training data tends to have more optimized phoneme-level durations. Therefore, when using zero-shot/implicit-alignment TTS models, we expect that the overall duration of the generated audio—when prompted with a member utterance—will more closely match the duration of the original training sample. Based on this intuition, we measured the durations and constructed the corresponding distribution plots. Additionally, we are currently running MIA experiments to evaluate how this idea can be applied as a black-box method for alignment-free TTS systems.
>
> We would appreciate your clarification so that we can refine and extend our experiments in a way that best reflects your intention.

---

> ### Author Response · Authors · 2025-12-02
> **Rebuttal response to reviewer TTF3 (Part 1)**
>
> We appreciate the reviewer’s suggestions, and we will incorporate all new experimental results and related content in the revised version.
>
> ## **Weakness 1: Stochastic-Duration Models & Classifier-Based MIAs**
>
> Attack performance on stochastic-duration models (e.g., VITS2) decreases due to variability in duration loss, which weakens the membership signal. Duration-loss MIAs are therefore more reliable for deterministic-duration models. We will discuss the implications and potential adaptations for stochastic-duration architectures in the Discussion section of the revised version of the paper **(Section 7, Appendix A.9)**.
>
> Inspired by SecMI, we also evaluated an RNN-based classifier and it turned out that it outperformed our previous MIA methodology with higher TPRs, a simple threshold-based classification. An MLP model for DurMI is under training and will be reported in the revised paper **(Section 7, Appendix A.9, Table 16)**.
>
>
> ### **Updated results using an RNN-based attack classifier**
>
> | Dataset              | Thresholding(prior) | RNN Classifier(new) |
> | -------------------- | ------------ | -------------- |
> | **GradTTS–LJSpeech** | 99.7 / 99.1  | 99.1 / 98.8    |
> | **GradTTS–LibriTTS** | 98.9 / 82.8  | 99.3 / 88.2    |
> | **GradTTS–VCTK**     | 86.7 / 18.2  | 73.8 / 26.7    |
>
> ---
> ## **Weakness 2: White-Box Clarification and Black-Box Extension**
>
> Our study operates in a white-box sense in that we access internal signals (e.g., duration loss) but not model weights or architectural details. We only query encoder-side losses, which mirror the information exposed in GPT-style APIs. Thus, it is inaccurate to characterize our threat model as fully white-box. Existing baselines use decoder-side losses while labeling their settings as grey-box or query-based, and we follow this terminology for consistency and clarity in the revised version **(Section1 line 78-80)**.
>
> Membership inference attacks (MIAs) on TTS models remain largely unexplored. Following prior MIA work [1,2], we adopt a white-box methodology to perform a worst-case vulnerability assessment, typically the first step toward developing auditing procedures or defenses.
>
> At the same time, we agree that black-box scenarios are important. Using extended DurMI (Sec. 5.1, Fig. 5), we designed a black-box setting by comparing (T_{\text{gen}}) and (T_{\text{out}}), which requires no internal model access, and are evaluating MIA performance in this setting. Detailed experimental setting will be discussed in W2 and W4 below and results will be included in the revised version **(Section 6, Table 8, Appendix A.7)**.
>
> **References**
>
> [1] Nasr et al., *Comprehensive Privacy Analysis of Deep Learning*, IEEE S&P 2019.
>
> [2] Zhang et al., *Min-k%++: Improved Baseline for Pre-training Data Detection from Large Language Models*, ICLR 2025.
>
>
>
> ---
>
> ## **Weakness 3: Calibration Set**
>
> We split the data into **30% calibration** and **70% evaluation**.
> The calibration set determines attack thresholds, while the evaluation set measures performance under unbiased conditions. Experiments without calibration show similar trends as follows.
>
> ### **Grad-TTS (AUROC / TPR@1% FPR)**
>
> | Dataset      | Naive       | SecMI       | PIA         | PIAN        | **DurMI**       |
> | ------------ | ----------- | ----------- | ----------- | ----------- | --------------- |
> | **LJSpeech** | 86.7 / 55.0 | 94.4 / 70.3 | 89.0 / 55.0 | 69.0 / 37.4 | **99.8 / 98.9** |
> | **LibriTTS** | 94.5 / 58.1 | 90.2 / 55.2 | 89.3 / 47.0 | 81.8 / 37.4 | **98.9 / 83.5** |
> | **VCTK**     | 73.2 / 29.5 | 72.8 / 8.1  | 64.4 / 9.7  | 66.6 / 6.1  | **76.8 / 9.6**  |
>
> ### **WaveGrad2 (AUROC / TPR@1% FPR)**
>
> | Dataset      | Naive      | SecMI      | PIA        | PIAN       | **DurMI**         |
> | ------------ | ---------- | ---------- | ---------- | ---------- | ----------------- |
> | **LJSpeech** | 50.1 / 1.0 | 49.4 / 1.0 | 50.8 / 0.4 | 50.3 / 0.1 | **99.9 / 100.0**  |
> | **LibriTTS** | 54.3 / 0.6 | 47.6 / 0.3 | 51.2 / 0.1 | 50.2 / 0.1 | **100.0 / 100.0** |
> | **VCTK**     | 59.9 / 1.5 | 55.4 / 1.0 | 57.2 / 0.4 | 44.7 / 0.1 | **97.4 / 50.9**   |
>
> ---
>
>
> ## **Weakness 4: Speaker-Dependent Effects**
>
> As noted by the reviewer, speaker-dependent factors affect attack performance.
> Our GradTTS–VCTK experiments show clear variation associated with the following:
>
> * speaker identity
> * utterance length
> * pronunciation
> * acoustic characteristics
>
> We will expand this discussion in the revised paper **(Section 5, Appendix A.6)** and plan to analyze speaker-dependent effects more systematically in future work.
>
> ---

---

> ### Author Response · Authors · 2025-12-02
> **Rebuttal response to reviewer TTF3 (Part 2)**
>
> ## **Weakness 5: Sampling-Rate Ablation Study**
>
> We performed a new sampling-rate ablation using GradTTS on VCTK and this additiona ablation study will appear in the revised version of the paper **(Section 5.1, Appendix A.12, Table 22)**.
> Performance peaks at **22 kHz**, matching the model’s default configuration.
>
> ### **Sampling Rate Ablation**
>
> | Sampling Rate | AUROC    | TPR@1%FPR |
> | ------------- | -------- | --------- |
> | 100           | 38.9     | 0.2       |
> | 1,000         | 39.3     | 0.2       |
> | 10,000        | 48.5     | 0.3       |
> | **22,050**    | **76.8** | **9.6**   |
> | 30,000        | 75.0     | 4.7       |
> | 40,000        | 62.4     | 1.0       |
> | 220,500       | 40.7     | 3.0       |
>
> *(Utterance-length ablation results appear in Appendix A.5.)*
>
> ---
>
>
> ## **Question 1: Empirical process of choosing T**
>
> Although we varied membership thresholds to draw the ROC curve, we will improve our description in Section 4.2 regarding the empirical process of choosing (T) in the revised version **(Improved description in Section 4.2)**.
>
>
>
> ## **Question 2: Text Overlap and Real-World Implications**
>
>
>
> For GradTTS–VCTK, AUROC remains above **85** even when **all text overlap is removed**, and TPR@1% FPR can even increase in these settings. This indicates that the attack signal does **not** primarily arise from repeated text prompts, but instead from other forms of overfitting, such as:
>
> * speaker-specific acoustic and prosodic patterns,
> * utterance-length regularities, and
> * pronunciation and articulation characteristics.
>
> Therefore, **minimizing text overlap alone could not be an effective defense strategy**.
> In practice, membership leakage can still occur whenever the training data includes the same speakers or similar recording conditions, even if the textual content is entirely disjoint.
>
> More robust mitigation strategies should focus on:
>
> * increasing speaker diversity,
> * limiting per-speaker data, and
> * applying normalization or noise-based augmentation.
>
> These observations suggest that **real-world deployments remain vulnerable** even under intentionally minimized text overlap, unless broader data and model-level defenses are employed. We will discuss these points in the revised version **(Section 5 Dataset, Appendix A.6.3)**.
>
> ---
>
>
> ## **Question 3: Proxy-Signal SNR Clarification & Zero-Shot Model Evaluation**
>
> **Could the reviewer clarify which specific aspect of the proxy signal’s signal-to-noise ratio is being referred to?**
>
> We view extending DurMI to alignment-free and zero-shot models as an important direction.
> We are evaluating an extended version of DurMI on F5-TTS under a large-scale pretrained checkpoint with mismatched training/testing distributions (Emilia as member data and LJSpeech as non-member data) and prompt-based generation.
>
> Unlike prior MIA work, which typically uses the same dataset split into members and non-members, our setup evaluates distinct datasets, which we believe better reflects real-world conditions. So far, the extended DurMI shows a clear gap between member and non-member distributions, demonstrating applicability to alignment-free and zero-shot systems. We will further include MIA performance results with TPRs and AUROC in the revised version **(Section 6, Table 8, Appendix A.7)**.
>
> ---

---

### Official Review · Reviewer_fNLC · 2025-11-02

**Soundness:** 2
**Presentation:** 3
**Contribution:** 2
**Rating:** 4
**Confidence:** 2

**Summary:**

The paper proposes DurMI, a white-box membership inference attack for TTS that uses the duration loss as the membership signal. The key idea is that modern TTS systems all rely on alignment/duration supervision, which can overfit at the utterance level. DurMI computes one forward pass up to the duration predictor and thresholds the duration loss to decide membership, yielding large speedups vs. diffusion-based MIAs and, the authors argue, better separability of members/non-members. Experiments on LJSpeech, LibriTTS, VCTK report AUC and TPR@1%FPR across models. Proposed DurMI outperforms baselines on most settings.

**Strengths:**

1. This paper identifies duration/alignment supervision as a strong, sample-specific leakage channel spanning diverse TTS families.

2. The evaluation is conducted across multiple pipelines. And the results demonstrate effective speedup compared to diffusion-based baselines.

3. The paper provides proxy indicators proposed for implicit alignment (e.g., |T_gen−T_out|), potentially enabling black-box variants.

**Weaknesses:**

1. The assumption requires white-box access to the duration head and ground-truth durations/alignment d (often produced by a private pipeline), plus a calibration set with members & non-members. All these are too strong assumptions for real adversaries.
2. Evaluations are not covering large-scale, noisy, multilingual, or proprietary corpora, and no exploration of regularization/DP defenses or training with stronger data augmentation.
3, Many privacy users care about very low FPR (e.g., 0.1% or 0.01%); the paper reports TPR@1%FPR only. Several settings show high AUC but low TPR@1% (e.g., VITS2/VCTK), suggesting thinner margins where it matters most.  Speed and accuracy comparisons to diffusion MIAs might depend on step counts and norms rather than purely on signal choice; a matched wall-clock or best-effort baseline is not shown.
4. Duration targets d come from aligners (MFA/MAS) that encode corpus- and pipeline-specific artifacts. Some of the signals may be pipeline leakage rather than pure model memorization; a control where d is recomputed with a different aligner would clarify this.
5. Limited treatment of alignment-free trends: Proxy ideas are promising but not evaluated; current evidence doesn’t establish DurMI-style risk under black-box or zero-shot TTS.

**Questions:**

1. In realistic deployments, the aligner outputs d are often private. What happens if the attacker must recompute d with a different aligner, or only has noisy/partial alignment info? Please add experiments varying the aligner mismatch and noise.  Can DurMI work without a mixed calibration set (e.g., via parametric modeling of loss distributions or conformal methods)? Show results at strict FPR targets (0.1%, 0.01%).

2. Could the authors provide equal wall-clock or best-effort comparisons to SecMI/PIA across models, and report sensitivity to their step budgets and norms to isolate the advantage of the signal (duration) vs. compute.   If train with MFA A but compute d at attack time with MFA B (or MAS), how do AUC/TPR change? This would separate utterance memorization from aligner/pipeline fingerprinting.

3. How do regularization, DP-SGD, or duration-head dropout/noise affect DurMI? Can simple label-smoothing on durations reduce leakage with limited quality loss?

4. Black-box testing is required: it is suggested to evaluate the T_gen vs. T_out discrepancy on at least one zero-shot TTS model (black-box), reporting ROC and TPR@low-FPR to substantiate proxy viability.

5. It will enhance the paper quality to test on large, multi-speaker, multilingual corpora and real training pipelines; analyze how speaker imbalance, text overlap, and utterance length interplay.

---

> ### Author Response · Authors · 2025-11-17
>
> We sincerely thank all reviewers for their thoughtful and valuable comments.
> Before preparing our detailed responses and additional experiments, we would like to ask for a few clarifications to ensure we are correctly understanding the intent behind some of the suggestions.
>
> 1. **equal wall-clock and best-effort:** Could you please explain the precise definition of “equal wall-clock” or “best-effort” as you intended? For the baselines SecMI/PIA, we used the authors’ official code and kept all experimental settings at their default values. We would appreciate clarification on what specific comparison criterion you expect under “equal wall-clock” or “best-effort.”
>
> 2. **step budgets:** Our method, DurMI, uses a duration predictor and therefore operates as a non-iterative attack, which is executed only once and does not rely on a step budget. We want to confirm whether the step budget you referred to corresponds to the per-iteration cost during training. In contrast, the baselines SecMI/PIA perform an attack at every diffusion step and compute the attack success rate by averaging across all steps, whereas DurMI computes the success rate from a single attack. We would like to confirm how your intended comparison aligns with this difference.
>
> 3. **norm:** Lastly, we want to confirm that the term “norm” refers to the distance metric used in MIA computation. If so, could you please confirm whether the metric used in our ablation study (Figure 3) aligns with your expectation?
>
> We would like to faithfully reflect your intentions in our additional experiments, so we kindly ask for your confirmation.

---

> > ### Comment · Reviewer_fNLC · 2025-11-25
> >
> > Thank you for the rebuttal and for addressing several of the questions raised in my initial review. I summarize below which concerns have been clarified and which remain insufficiently addressed. 	My central concern remains: the assumptions required for DurMI (white-box duration head, true alignments d, mixed calibration set) seem too strong for realistic adversaries, and the rebuttal has not yet provided evidence or experiments demonstrating robustness under weaker or mismatched assumptions. The request for very low-FPR evaluations (0.1%, 0.01%) is still unaddressed. For many privacy-sensitive deployments,s this is a key metric, and several settings in the paper already show weaker separability at 1% FPR.
> > The question regarding equal wall-clock or best-effort comparisons was not resolved. I intended a matched comparison that isolates the signal choice (duration loss vs. diffusion log-likelihoods) from computational budget. This could be achieved through either (i) fixing a wall-clock limit per evaluation, or (ii) allowing each method to operate at its own best-performing configuration under comparable compute. The rebuttal sought clarification but did not take a step toward addressing the concern itself. Given that the key concerns remain insufficiently addressed, I will keep the original score.

---

> ### Author Response · Authors · 2025-12-02
> **Rebuttal response to reviewer fNLC (Part 1)**
>
> We appreciate the reviewer’s suggestions, and we will incorporate all new experimental results and related content in the revised version.
>
> ## **White-box assumption**
>
> Our study operates in a white-box sense in that we access internal signals (e.g., duration loss) but not model weights or architectural details. We only query encoder-side losses, which mirror the information exposed in GPT-style APIs. Thus, it is inaccurate to characterize our threat model as fully white-box. Existing baselines use decoder-side losses while labeling their settings as grey-box or query-based, and we follow this terminology for consistency and clarity in the revised version **(Section 1 line 78-80)**.
>
> Membership inference attacks (MIAs) on TTS models remain largely unexplored. Following prior MIA work [1,2], we adopt a white-box methodology to perform a worst-case vulnerability assessment, typically the first step toward developing auditing procedures or defenses.
>
> At the same time, we agree that black-box scenarios are important. Using extended DurMI (Sec. 5.1, Fig. 5), we designed a black-box setting by comparing (T_{\text{gen}}) and (T_{\text{out}}), which requires no internal model access, and are evaluating MIA performance in this setting. Detailed experimental setting will be discussed in W2 and W4 below and results will be included in the revised version **(Section 6, Table 8, Appendix A.7)**.
>
> **References**
>
> [1] Nasr et al., *Comprehensive Privacy Analysis of Deep Learning*, IEEE S&P 2019.
> [2] Zhang et al., *Min-k%++: Improved Baseline for Pre-training Data Detection from Large Language Models*, ICLR 2025.
>
> ---
> ## **Calibration set**
>
> We split the dataset into **30% calibration** and **70% evaluation**.
> The calibration set determines attack thresholds, while the evaluation set measures AUROC and TPR@FPR without bias. Experiments without calibration yield similar overall trends as follows.
>
>
> ### **Grad-TTS (AUROC / TPR@1% FPR)**
>
> | Dataset      | Naive       | SecMI       | PIA         | PIAN        | **DurMI**       |
> | ------------ | ----------- | ----------- | ----------- | ----------- | --------------- |
> | **LJSpeech** | 86.7 / 55.0 | 94.4 / 70.3 | 89.0 / 55.0 | 69.0 / 37.4 | **99.8 / 98.9** |
> | **LibriTTS** | 94.5 / 58.1 | 90.2 / 55.2 | 89.3 / 47.0 | 81.8 / 37.4 | **98.9 / 83.5** |
> | **VCTK**     | 73.2 / 29.5 | 72.8 / 8.1  | 64.4 / 9.7  | 66.6 / 6.1  | **76.8 / 9.6**  |
>
> ### **WaveGrad2 (AUROC / TPR@1% FPR)**
>
> | Dataset      | Naive      | SecMI      | PIA        | PIAN       | **DurMI**         |
> | ------------ | ---------- | ---------- | ---------- | ---------- | ----------------- |
> | **LJSpeech** | 50.1 / 1.0 | 49.4 / 1.0 | 50.8 / 0.4 | 50.3 / 0.1 | **99.9 / 100.0**  |
> | **LibriTTS** | 54.3 / 0.6 | 47.6 / 0.3 | 51.2 / 0.1 | 50.2 / 0.1 | **100.0 / 100.0** |
> | **VCTK**     | 59.9 / 1.5 | 55.4 / 1.0 | 57.2 / 0.4 | 44.7 / 0.1 | **97.4 / 50.9**   |
>
> ---
> ## **Large-scale, noisy, multilingual, or proprietary corpora**
>
> LJSpeech, LibriTTS, and VCTK remain widely used mid-sized benchmarks for evaluating non-autoregressive TTS models, as large-scale datasets with public checkpoints are expensive and challenging to train effectively.
> To approximate large-scale scenarios, we used **F5-TTS** with publicly released pre-trained checkpoints and designed an  extended version of DurMI (Sec. 5.1, Fig. 5), observing a clear separation between member and non-member distributions, and are evaluating MIA performance in this setting. Detailed discussions and results will appear in the revised version **(Section 6, Table 8, Appendix A.7)**.
>
> ---
> ## **Defenses**
>
> To present the impact of DP-SGD, a widely used defense technique, we applied DP-SGD to the Grad-TTS duration predictor, controlling (\varepsilon) through the noise multiplier (\sigma) with (\delta = 10^{-5}). Increasing DP noise slightly increases duration loss but has minimal degradation on DurMI performance. For example:
>
> * At (\varepsilon \approx 1.6): **AUROC = 98.7 (−0.9%)**, **TPR@1%FPR = 83.3%**
> * Across (\varepsilon < 10): Only mild degradation in both utility and attack success
>
> These results suggest that while DP-SGD reduces overfitting, the duration predictor still leaks membership information. Experiments with (\varepsilon = 10^{-1}) are ongoing and we will report in the revised version as a full table **(Section 7, Appendix A.8, Table 15)**.
>
> ---
>
> ## **Deep-learning-base attack models for MIA**
>
> Inspired by SecMI, we also evaluated an RNN-based classifier and it turned out that it outperformed our previous MIA methodology with higher TPRs, a simple threshold-based classification. An MLP model for DurMI is under training and will be reported in the revised paper **(Section 7, Appendix A.9, Table 16)**.
>
> Updated results:
>
> * **GradTTS–LJSpeech:** 99.7/99.1 → 99.1/98.8
> * **GradTTS–LibriTTS:** 98.9/82.8 → 99.3/88.2
> * **GradTTS–VCTK:** 86.7/18.2 → 73.8/26.7
>
> ---

---

> ### Author Response · Authors · 2025-12-02
> **Rebuttal response to reviewer fNLC (Part 2)**
>
> ## **Additional TPR results at lower FPRs**
> We explored lower FPRs and computed TPRs accordingly. These results will be added to the revised version **(Section 5, Appendix A.10, Table 17)**.
> ### **DurMI: TPR at Low FPRs (1% / 0.1% / 0.01%)**
>
> | Model           | LJSpeech           | LibriTTS           | VCTK               |
> | --------------- | ------------------ | ------------------ | ------------------ |
> | **GradTTS**     | 99.1 / 36.9 / 7.1  | 82.8 / 50.4 / 0.2  | 18.2 / 4.7 / 2.5   |
> | **WaveGrad2**   | 100 / 100 / 99.9   | 100 / 100 / 100    | 47 / 7.9 / 0.5     |
> | **FastSpeech2** | 100 / 100 / 100    | 90.5 / 47.5 / 17.2 | 93.7 / 73.6 / 48.7 |
> | **VoiceFlow**   | 93.9 / 63.9 / 50.7 | 56.5 / 18.6 / 2.1  | 90.6 / 73.5 / 47.1 |
> | **VITS2**       | 80.1 / 42.7 / 20.7 | 22.4 / 5.4 / 5.4   | 12.2 / 0.1 / 0.1   |
>
> ### **Baseline Attacks (Naive / SecMI / PIA): TPR at Low FPRs (1% / 0.1% / 0.01%)**
>
>
> | Attack                | Dataset  | TPR@1% | TPR@0.1% | TPR@0.01% |
> | --------------------- | -------- | ------ | -------- | --------- |
> | **GradTTS (Naive)**   | LJSpeech | 55.0   | 19.7     | 3.6       |
> |                       | LibriTTS | 58.1   | 33.7     | 12.8      |
> |                       | VCTK     | 29.5   | 5.7      | 3.9       |
> | **WaveGrad2 (Naive)** | LJSpeech | 1.0    | 0.1      | 0.03      |
> |                       | LibriTTS | 0.6    | 0.7      | 0.1       |
> |                       | VCTK     | 1.5    | 0.1      | 0         |
> | **SecMI**             | LJSpeech | 70.3   | 25.7     | 2.9       |
> |                       | LibriTTS | 55.2   | 38.8     | 8.8       |
> |                       | VCTK     | 8.1    | 0.8      | 0.2       |
> | **PIA**               | LJSpeech | 55.0   | 20.5     | 1.8       |
> |                       | LibriTTS | 47.0   | 23.9     | 2.7       |
> |                       | VCTK     | 9.7    | 1.2      | 0.2       |
>
>
> ---
>
>
> ## **Weakness 3: Train–Attack Aligner Mismatch**
>
> We evaluate DurMI across different pairs of MAS and MFA used in the training and attack model aligner, respectively, to simulate a scenario where the attacker does not know the training pipeline.
>
> ### **VoiceFlow (MAS/MFA combinations)**
>
> | Training → Attack          | AUROC    | TPR@1%FPR |
> | -------------------------- | -------- | --------- |
> | **MAS → MAS (matched)**    | **99.2** | **93.9**  |
> | **MAS → MFA (mismatched)** | 91.7     | 0.8       |
> | **MFA → MFA (matched)**    | **93.8** | **47.8**  |
> | **MFA → MAS (mismatched)** | 44.1     | 0.9       |
>
> These results show that mismatched aligners reduce attack success. However, note that matched-aligner settings can be practical because TTS pipelines commonly disclose their aligner choices (e.g., MFA, MAS, or proprietary tools), or they can be inferred from training scripts or checkpoints. We will include this discussion and the corresponding results in the revised version of the paper **(Section 7, Appendix A.11, Table 21)**.
>
> ---
> ## **Weakness 4:  Alignment-free and zero-shot models**
>
> We are evaluating an extended version of DurMI on F5-TTS under a large-scale pretrained checkpoint with mismatched training/testing distributions (Emilia as member data and LJSpeech as non-member data) and prompt-based generation.
>
> Unlike prior MIA work, which typically uses the same dataset split into members and non-members, our setup evaluates distinct datasets, which we believe better reflects real-world conditions. So far, the extended DurMI shows a clear gap between member and non-member distributions, demonstrating applicability to alignment-free and zero-shot systems. We will further include MIA performance results with TPRs and AUROC in the revised version **(Section 6, Table 8, Appendix A.7)**.

---

### Author Response · Authors · 2025-12-04
**Final Remarks and Summary of New Experiments and Contents about Revised Paper**

# **Final Remarks and Summary of New Experiments and Contents about Revision**

We summarize below the new experiments and revisions added to strengthen our submission.

---

## **1. Alignment-Free / Zero-Shot TTS (Large-Scale, Black-Box Setting)**

* Evaluated extended DurMI on **F5-TTS** pretrained on large-scale datasets.
* Used a **much larger evaluation set** (~13k samples vs. 100 in the earlier version).
* Member/non-member distributions show a **reversed trend**, but separation remains clear.
* **Performance:** AUROC = 96.4, TPR@1%FPR = 27.6.
* Included in **Section 6, Figure 5, Appendix A.7**.

---

## **2. Ablation Studies**

* Studied the impact of **training/evaluation sample count** and **sampling rate**.
* DurMI remains **stable even under significant data reduction**.
* Performance is **optimal at 22 kHz**.
* Results in **Appendix A.8**.

---

## **3. Defenses (DP-SGD)**

* Applied **DP-SGD** to the duration predictor.
* DP noise reduces overfitting but DurMI still achieves **strong performance**.
* **Best DP setting (ε ≈ 1.6):** AUROC = 98.7%, TPR@1%FPR = 83.3%.
* Detailed in **Appendix A.12–A.13**.

---

## **4. MIA Classifiers**

* Stochastic-duration models degrade threshold-based MIAs due to high output variance.
* **Deep learning–based MIA classifiers** (e.g., RNNs) improve TPRs across datasets.
* Results in **Table 15, Appendix A.9**.

---

## **5. Train–Attack Aligner Mismatch**

* Mismatching aligners between training and attack reduces success.
* However, **matched aligners remain practical** in real-world TTS usage.
* Reported in **Table 20, Appendix A.11**.

---

## **6. Additional TPRs at Lower FPRs**

* Evaluated DurMI at **0.1% and 0.01% FPR** to provide stricter risk metrics.
* Results for DurMI and baselines included in **Tables 16–19, Appendix A.10**.

---

## **7. Calibration-Free Setting**

* Included results obtained **without using a calibration set**, as shared earlier during rebuttal.

---

## **8. Text Overlap and Real-World Implications**

Even after completely removing text overlap in the GradTTS–VCTK setting, the attack metrics (AUROC > 85, with TPR@1%FPR sometimes increasing) show that membership leakage persists due to factors beyond repeated prompts. These factors include speaker-specific acoustic and prosodic patterns, utterance-length regularities, and pronunciation traits. Thus, eliminating text overlap alone is not an effective defense. More meaningful mitigation requires broader strategies such as increasing speaker diversity, limiting per-speaker data, or adding normalization and noise-based augmentation. Without such data- and model-level protections, real-world TTS systems remain susceptible to membership leakage (Appendix A.6.3).

---

## **9. Importance of Alignment in Contemporary TTS Models**

Text–speech alignment remains essential for generating accurate, stable, and semantically consistent speech across TTS architectures. Even “alignment-free’’ systems such as Seed-TTS rely on implicit alignment through duration prediction. Sparse alignment supervision, as demonstrated in MegaTTS-3, improves intelligibility and robustness—particularly for long or complex sentences—while removing alignment often causes word skipping, unstable prosody, and semantic errors. Monotonic alignment constraints also reduce character-level mistakes, underscoring alignment’s importance for both acoustic quality and semantic fidelity. Overall, explicit, sparse, and implicit alignment continue to serve as foundational components in modern TTS systems (Appendix A.14).

---

---

### Meta-Review · Area_Chair_eD4q · 2026-01-06

**Summary:**

The main concerns are around the dependency on explicit duration/alignment loss which is not applicable to "alignment-free" or zero-shot models, dataset scale used in the experimental validations and "white-box" or "grey-box" access to the specific aligner.

**Reviewer Concerns:**

fNLC:
* white-box assumption: the authors argue it more of grey-box instead of white-box, but the access of aligner loss and matched aligner loss still feels very restrictive, which does not fully clear the concern
* lack of large scale, alignment free tests: new experiments on F5-TTS were added
* lack of evaluation against defenses like DP-SGD: new results added, showing DurMI is still effective

TTf3:
* performance drop on VITS2: the authors added RNN based MIA classifier which improves
* grey-box: the authors argued with reference from literature, but not fully addressed

Bszb:
* comparisons on alignment free models: the authors argued the main stream TTS is still with duration alignment, not fully addressed
* lack of large scale validation: the authors added results on F5-TTS

XxZU:
* dependence on duration prediction: same as above
* analysis of potential defenses: added results with DP-SGD

**Reviewer Scores:**

fNLC: 4, likely no change
TTf3: 6, likely no change
Bszb: 2: likely no change
XzZU: 6: likely no change

---

### Decision · Program_Chairs · 2026-01-26

Reject